# Evaluation of Electrochemotherapy with Bleomycin in the Treatment of Colorectal Hepatic Metastases in a Rat Model

**DOI:** 10.3390/cancers15051598

**Published:** 2023-03-04

**Authors:** Antonios E. Spiliotis, Sebastian Holländer, Jeannette Rudzitis-Auth, Gudrun Wagenpfeil, Robert Eisele, Spyridon Nika, Orestis Mallis Kyriakides, Matthias W. Laschke, Michael D. Menger, Matthias Glanemann, Gereon Gäbelein

**Affiliations:** 1Department of Surgery, Campus Charité Mitte, Campus Virchow Klinikum, Charité—Universitätsmedizin Berlin, Corporate Member of Freie Universität Berlin and Humboldt-Universität zu Berlin, Augustenburger Platz 1, 13353 Berlin, Germany; 2Department of General Surgery, Vascular-, Visceral- and Pediatric Surgery, Saarland University Medical Center, 66421 Homburg, Germany; 3Institute for Clinical and Experimental Surgery, Saarland University, 66421 Homburg, Germany; 4Institute for Medical Biometry, Epidemiology and Medical Informatics, Saarland University Medical Center, 66421 Homburg, Germany; 5Department of Urology and Pediatric Urology, Saarland University Medical Center, 66421 Homburg, Germany

**Keywords:** electrochemotherapy, electroporation, bleomycin, liver neoplasms, rats

## Abstract

**Simple Summary:**

The available ablative procedures for the treatment of hepatic cancer have contraindications due to the heat-sink effect and the risk of thermal injuries. Electrochemotherapy (ECT) represents an ablative procedure that combines the administration of chemotherapeutic agents with well-dosed electric pulses for cell membrane reversible electroporation (rEP). In ECT, the enhanced cellular permeability facilitates the transportation of chemotherapeutic agents into tumor cells. In our study, ECT was compared with rEP and chemotherapy in a rat liver metastasis model. The ECT group showed a stronger reduction in tumor oxygenation compared to the rEP and chemotherapy groups. Histological analyses revealed a significantly increased tumor necrosis of >85% and a reduced tumor vascularization in the ECT group. Consequently, ECT is an effective treatment of hepatic tumors and may be utilized for the treatment of tumors adjacent to high-risk regions, where other ablative procedures are contraindicated.

**Abstract:**

Background: The available ablative procedures for the treatment of hepatic cancer have contraindications due to the heat-sink effect and the risk of thermal injuries. Electrochemotherapy (ECT) as a nonthermal approach may be utilized for the treatment of tumors adjacent to high-risk regions. We evaluated the effectiveness of ECT in a rat model. Methods: WAG/Rij rats were randomized to four groups and underwent ECT, reversible electroporation (rEP), or intravenous injection of bleomycin (BLM) eight days after subcapsular hepatic tumor implantation. The fourth group served as Sham. Tumor volume and oxygenation were measured before and five days after the treatment using ultrasound and photoacoustic imaging; thereafter, liver and tumor tissue were additionally analysed by histology and immunohistochemistry. Results: The ECT group showed a stronger reduction in tumor oxygenation compared to the rEP and BLM groups; moreover, ECT-treated tumors exhibited the lowest levels of hemoglobin concentration compared to the other groups. Histological analyses further revealed a significantly increased tumor necrosis of >85% and a reduced tumor vascularization in the ECT group compared to the rEP, BLM, and Sham groups. Conclusion: ECT is an effective approach for the treatment of hepatic tumors with necrosis rates >85% five days following treatment.

## 1. Introduction

Over the past 30 years, several ablative methods have been developed for the treatment of hepatic cancer as an alternative to surgical resection and liver transplantation, mainly in patients with unresectable cancer or in selected cases with resectable disease [1,2,3]. Recent studies and guidelines recommend radiofrequency ablation (RFA) and microwave ablation (MWA) as the ablative methods with the highest efficacy [1,2,4,5]; however, both procedures have contraindications that limit their utilization [2,4]. Specifically, they are not indicated for tumors adjacent to major hepatic vessels or bile ducts due to the heat-sink effect, and tumors in the vicinity of other organs due to the risk of thermal injuries [1,2,4,6,7,8].

To overcome those limitations, new nonthermal ablation technologies have been developed based on the principle of electroporation (EP) of the cellular membrane through application of an external electric field. EP is defined as the phenomenon that occurs when cells are exposed to a high external electric field. Through EP, a transmembrane voltage is induced on the cellular membrane that exceeds a threshold value, causing formation of hydrophilic pores and increased cellular permeability. Electric pulses can cause reversible (rEP) or irreversible EP. In reversible EP, pore formation and membrane destabilization are transient, and cells regain homeostasis [9]. On the other hand, in the irreversible EP, magnitude and duration of applied electrical pulses overwhelm the adaptive capacity of the cell membrane, causing irreversible injury and cell death [9,10].

Electrochemotherapy (ECT) represents an ablative procedure that combines the administration of chemotherapeutic agents with well-dosed electric pulses for cell membrane rEP. In ECT, the enhanced cellular permeability facilitates the transportation of otherwise poorly penetrating chemotherapeutic agents into tumor cells, increasing their cytotoxicity and inducing cell death [11,12].

ECT is a well-established method with palliative intent for the treatment of cutaneous and subcutaneous neoplastic lesions [11]. This procedure has been associated with beneficial oncological outcomes in the treatment of skin cancer with tumor response rates up to 86% [11,13,14] and has been included in the European guidelines for the treatment of advanced melanoma and primary squamous carcinoma [9,15].

The beneficial outcomes of ECT in the treatment of cutaneous cancer have also increased the interest in utilization of this method for nonsuperficial tumors. Potentially, ECT as a nonthermal approach could be utilized for the treatment of tumors adjacent to high-risk regions, where RFA and MWA are not indicated. Retrospective studies with a limited number of patients have been conducted to evaluate the role of ECT in liver tumors [16]; however, immunohistological examinations and photoacoustic imaging following ECT in liver tissue are poorly investigated as well as the role of the procedure in colorectal metastases. Therefore, we evaluated in the present study the effectiveness of ECT in comparison to chemotherapy and rEP as monotherapies in a rat liver metastasis model.

## 2. Materials and Methods

### 2.1. The Animals

Thirty-two WAG/Rij rats of both gender (males n = 16, females n = 16) with a body weight of 265.0 ± 14.5 g and an age of 46.0 ± 1.9 weeks were used for the experiments (Institute for Clinical and Experimental Surgery, Saarland University, Homburg/Saar, Germany). The animals were housed in groups and in a temperature- and humidity-controlled 12 h light/dark cycle environment with free access to water and standard laboratory chow.

The study was performed in accordance with the European legislation on the protection of animals (Directive 2010/63/EU) and the National Institutes of Health guidelines for the Care and Use of Laboratory Animals [17]. All experiments were authorized by the local governmental animal protection committee (Landesamt für Verbraucherschutz, Saarbrücken, Germany; permission number: 21/2019).

### 2.2. Experimental Protocol

The rats were randomized into four groups (n = 8 per group; 4 males and 4 females). All animals underwent laparotomy with tumor cell implantation in the left liver lobe (day 0). On day 8, relaparotomy was performed and all animals underwent ultrasound and photoacoustic imaging. Based on the treatment, the animals were divided into four groups. The first group underwent ECT with intravenous administration of bleomycin (BLM), the second group received only systemic chemotherapy with BLM, the third group underwent rEP with intravenous injection of an equivalent amount of 0.9% saline solution (B. Braun Melsungen AG, Melsungen, Germany), and the fourth group underwent laparotomy and surgical exposure of the liver without treatment (Sham). BLM was used in the experiments, as it has been proven that among other tested chemotherapeutic agents BLM has the highest potentiation of cytotoxicity by rEP [9,12,18].

Five days following the treatment (day 13), the animals underwent relaparotomy for the final ultrasound and photoacoustic imaging and for the collection of venous blood samples. Thereafter, the animals were sacrificed by an intravenous overdose of sodium pentobarbital and samples of the left liver lobe, including tumor and normal hepatic tissue, were asserved for histological and immunohistochemical analyses. The body weight of the animals was measured on days 0, 8, and 13 to assess eventual weight loss due to the treatment.

### 2.3. Tumor Cell Implantation

CC531 rat colon carcinoma cells (CLS, Heidelberg, Germany), which are syngeneic to WAG/Rij rats, were cultured as described in previously published studies [19]. For the induction of colorectal liver metastases, the rats were positioned in supine position on an electronically regulated heating pad, which adjusted the body temperature at 37 °C. Under isoflurane anesthesia, we conducted a median laparotomy and subcapsular implantation of syngeneic 5 × 10^5^ CC531 cells (in 50 µL phosphate buffered saline) on the lower surface of the left liver lobe using a 30G × 1/2” needle (Omnican^®^ F, B. Braun Melsungen AG, Melsungen, Germany). Then, the liver was repositioned anatomically into the peritoneal cavity and the laparotomy was closed with a one-layer running Vicryl 4-0 suture (Ethicon/Johnson & Johnson Medical Ltd., Livingston, UK).

### 2.4. ECT

Animals were anesthetized by isoflurane inhalation and were positioned in supine position on an electronically regulated heating pad. After median laparotomy, the inferior vena cava was exposed, and BLM (BLEO-cell^®^ 15 mg, Stadapharm GmbH, Bad Vilbel, Germany) was administrated as an intravenous bolus injection at a concentration of 4 U/kg body weight with a 30G × 1/2” needle (BD Microlance^TM^ 3, Becton Dickinson GmbH, Drogheda, Ireland) according to previously published standards [20]. Thereafter, the liver was mobilized, and the left liver lobe was exposed to facilitate the electrodes’ insertion. At the timepoint of treatment, the tumors exhibited a mean diameter of 5 mm, as measured by ultrasound imaging, and presented as spherically symmetrical or asymmetrical lesions. Two parallel needle electrodes in a fixed geometry with inter-electrode distance of 8 mm, length of 20 mm, and diameter of 2.36 mm were used for the experiments. The electrodes were inserted in a plastic holder and connected to the electric pulse generator (Sennex^®^ Tumor System, BIONMED^®^ Technologies GmbH, Saarbrücken, Germany) (Figure 1). The parallel electrodes were placed around the external borders of the tumor at a depth of 5 mm to achieve sufficient electric field coverage for the ECT.

Eight electric pulses of 100 μs duration were applied between the two electrodes 3 min after the BLM injection [21]. According to the European guidelines for the utilization of ECT, the Sennex^®^ Tumor System provides a voltage between the pair of electrodes at 1.000 V, corresponding to an amplitude of 125 V/mm and frequency of 1 Hz [13,22]. After pulse delivery, the electrodes were removed from the liver parenchyma, and hemostasis was obtained by gauze packing. Then, the liver was repositioned anatomically into the peritoneal cavity and the laparotomy was closed again.

### 2.5. Ultrasound and Photoacoustic Imaging

A Vevo LAZR system (FUJIFILM VisualSonics Inc., Toronto, ON, Canada) and a real-time microvisualization LZ550 linear-array transducer (FUJIFILM VisualSonics Inc.) with a center frequency of 40 MHz were used for the ultrasound and photoacoustic imaging of the tumors.

The ultrasound examination was conducted during the laparotomy on day 8 before treatment and on day 13 before animals’ euthanasia. During the examination, the animals were anesthetized by isoflurane inhalation and were fixed on a heated stage. Heart and breathing rate were constantly monitored and the body temperature was maintained at 36–37 °C (THM150; Indus Instruments, Houston, TX, USA). A colorless aqueous warmed ultrasound gel (Aquasonic 100, Parker Laboratories, Inc., Fairfield, NJ, USA) without air bubbles was applied on the tumor surface to facilitate the efficient transduction of the photoacoustic and ultrasound signal.

The 40-MHz ultrasound probe was attached to a stepping motor that moved the probe over the surface of the left liver lobe. The ultrasound imaging was obtained through parallel two-dimensional images acquired in uniformly spaced intervals of 300 μm. The three-dimensional reconstruction of the tumor was achieved by off-line-outlining the tumor borders on every two-dimensional image. Utilizing these data, the integrated software of the Vevo LAZR system produced a polygonal three-dimensional image of the tumor and calculated the tumor volume.

OxyHemo-mode photoacoustic images were taken at two wavelengths (750 nm and 850 nm) with a two-dimensional photoacoustic gain of 42 dB and a hemoglobin threshold of 20 dB. Photoacoustic examination was performed at 750 nm and 850 nm wavelength illumination to deduce oxygen saturation (SO_2_) from the oxygenated and deoxygenated hemoglobin signal [23]. Furthermore, at these wavelengths any potential effect of edema caused by the ECT treatment on the photoacoustic signal was minimal due to low water absorption [24]. Hemoglobin concentration (HbT) and SO_2_ were calculated in the whole tumor tissue. All ultrasound images were examined independently by three researchers with experience in liver ultrasound imaging. The final evaluation for each animal was given after agreement of the three researchers.

### 2.6. Sampling and Assays

Via puncture of the subhepatic vena cava, venous blood samples were taken on day 13 directly after the ultrasound imaging. The blood cell analysis was conducted to evaluate the inflammatory reaction and the blood loss following the treatment. The total number of leukocytes (10^9^/L), lymphocytes (10^9^/L), monocytes (10^9^/L), neutrophils (10^9^/L), erythrocytes (10^12^/L), and platelets (10^9^/L) as well as the hemoglobin concentration (g/dL) and hematocrit (%) were assessed by means of a cell counter (VetScan HM5, Firma Scil Animal Care Company GmbH, Viernheim, Germany).

### 2.7. Histological and Immunohistochemical Analysis

The left liver lobe with the tumor and the surrounding normal hepatic parenchyma was fixed in 4% phosphate-buffered formalin, embedded in paraffin, and cut into 3 µm-thick sections. For the assessment of the necrotic tumor tissue, sections were stained with hematoxylin–eosin.

For the immunohistochemical detection of apoptotic cells, sections were stained with a rabbit polyclonal anti-cleaved caspase-3 antibody (1:100, Cell Signaling Technology, Frankfurt, Germany). For the streptavidin–biotin complex peroxidase staining, a biotinylated anti-rabbit Ig antibody served as secondary antibody (ready-to-use, Abcam, Cambridge, UK). For the immunohistochemical detection of proliferating cells, sections were stained with a monoclonal mouse-anti-human anti-proliferating cell nuclear antigen (PCNA) antibody (1:100; Dako, Hamburg, Germany). A peroxidase-conjugated goat anti-mouse IgG antibody (1:100; Dianova, Hamburg, Germany) served as secondary antibody. For the immunohistochemical detection of microvessels, sections were stained with a polyclonal rabbit CD31 antibody (1:200; Abcam, Cambridge, UK). A biotinylated goat anti-rabbit Ig antibody (ready-to-use, Abcam, Cambridge, UK) served as secondary antibody. For the immunohistochemical detection of neutrophilic granulocytes, sections were incubated with a rabbit polyclonal anti-myeloperoxidase (MPO) antibody (1:100; Abcam, Cambridge, UK) as primary antibody, followed by a biotinylated goat anti-rabbit IgG antibody (ready-to-use; Abcam, Cambridge, UK). 3-Amino-9-ethylcarbazole (Abcam, Cambridge, UK) or 3.3′-diamino-benzidine were used as chromogens, and Mayer’s hemalum (Merck, Darmstadt, Germany) served as counterstaining.

All histological and immunohistochemical analyses were conducted with the BX60 microscope (Olympus, Hamburg, Germany) and the imaging software cellSens Dimension 1.15 (Olympus, Hamburg, Germany). For this purpose, the sections were coded and microscopically analysed by three independent researchers, who were blinded with respect to the treatment group. Necrotic areas in the tumor tissue were measured as percentage of the whole tumor area on the section with the largest cross-sectional diameter of the tumor. For the assessment of cell proliferation, a semiquantitative index was developed, where every specimen was classified into one of five categories based on the percentage of PCNA-positive tumor cells relative to the total number of tumor cells in 10 high power fields (HPF) of non-necrotic tumor tissue (0 ≤ 1%, 1 = 1–10%, 2 = 11–30%, 3 = 31–50%, and 4 ≥ 50% of PCNA-positive cells). Cleaved caspase-3-positive cells, MPO-positive neutrophilic granulocytes, as well as CD31-positive microvessels were counted in 10 randomly selected HPF of non-necrotic tumor tissue (5 HPF in the center and 5 HPF in the periphery of the tumor). Positive cells were given as absolute number per HPF and the microvessel density was given in mm^−2^.

### 2.8. Statistical Analysis

All values are expressed as mean ± SEM (standard error of the mean). After analysis of the normal distribution of data and homogeneity of variance, differences between the groups were calculated by one-way analysis of variance (ANOVA). We did not account for the issue of multiple post-hoc testing due to the explorative nature of the investigation. The pairwise comparison was performed by student’s *t* test. Overall statistical significance is due to a two-sided significance level of 0.05. Statistical analysis was performed with the use of IBM-SPSS, version 28.0.1.0.

## 3. Results

### 3.1. Tumor Development and General Health Conditions

Eight days following tumor implantation, all animals developed a tumor in the left liver lobe with a mean diameter of 5 mm and a volume of 43.2 ± 4.1 mm^3^ (mean ± SEM). Accordingly, the animals in all groups exhibited a similar tumor growth without marked differences in tumor volume at the day of treatment (*p* = 0.590).

The assessment of body weight at regular intervals revealed a slight but not significant decrease from day 0 to day 13, which did not exceed 10% of the total body weight. Peritoneal or other extrahepatic metastases were not detected. The animals were not affected systemically by the malignant process and showed normal feeding and cleaning habits throughout the observation period.

### 3.2. Ultrasound and Photoacoustic Imaging

On day 8 before treatment, the ultrasound and photoacoustic imaging showed similar tumor volume (ECT 38.7 ± 6.2 mm^3^; rEP 44.8 ± 10.0 mm^3^; BLM 37.3 ± 2.5 mm^3^; and Sham 52 ± 11.4 mm^3^), SO_2_, and HbT among the groups. Five days following ECT, a 35.5% reduction in oxygenation was detected in the treated area (SO_2_ pretreatment: 72.6 ± 5.7%; SO_2_ posttreatment: 46.8 ± 5.5%). The rEP and BLM groups presented a slight reduction in tumor oxygenation, which was assessed at 17.3% and 22.8%, respectively (rEP pretreatment: 79.9 ± 1.7%; rEP posttreatment: 66.1 ± 6.6%; BLM pretreatment: 77.3 ± 3.0%; and BLM posttreatment: 59.7 ± 3.8%). The posttreatment oxygenation level in the ECT group was significantly lower compared to the rEP and Sham group, whereas compared to the BLM group the difference in oxygenation did not reach a significant level (Figure 2).

The ECT group demonstrated the lowest levels of HbT in the treated tissue compared to the other groups; furthermore, although the reduction in HbT after ECT was limited to 6.8% compared to the pretreatment values, only ECT-treated animals presented a reduction in hemoglobin levels. On the contrary, the hemoglobin levels in the rEP, BLM, and Sham groups were increased by 53.6%, 53.9%, and 47.3% compared to pretreatment values, respectively (Figure 2). Statistically significant differences in pre- and post-treatment values were only found in the Sham group.

### 3.3. Tumor Necrosis

Macroscopically, the ECT-treated areas were sharply demarcated from the normal hepatic parenchyma with oval or irregular shape. Microscopically, extensive necrosis was detected in the treated areas. A fibrotic zone was observed at the peripheral parts of the necrotic tumor tissue, which consisted of fibroblastic tissue, inflammatory cells (foamy macrophages, lymphocytes, plasma cells, eosinophils, and histiocytes), and proliferating capillaries with red blood cell (RBC) extravasates. Small numbers of residual tumor cells in combination with inflammatory cell infiltration were seen in the periphery of the treated area. The fibrotic pseudocapsule was surrounded by normal hepatic parenchyma and regenerative marginal hepatocytes. Beyond that region, hepatic parenchyma was viable without tumor infiltration. The wall of vessels and bile ducts adjacent to the ablation zone remained intact.

An ablated area was detected at the electrodes’ insertion site, which was oval with a diameter of 3–4 mm. In these foci, the hepatic tissue was necrotic, including perforated blood vessels or dilated capillaries with RBC extravasates, and was surrounded by inflammatory cells.

The histological analysis revealed a higher rate of tumor necrosis in the ECT group, which was estimated at 87.2 ± 3.1%; hence, ECT showed enhanced effectiveness compared to the other groups, where the tumor necrosis was lower. Tumors treated by BLM or rEP showed histological aspects indistinguishable from native nontreated tumors (Figure 3).

### 3.4. Apoptotic Cell Death

A limited number of apoptotic cells were observed in the periphery of the ablation zone among the residual tumor cells in the fibrotic pseudocapsule. The treatment with ECT was not associated with increased hepatocellular apoptotic death compared to the rEP and BLM groups (ECT 6.5 ± 2.3 per HPF; rEP 4.0 ± 1.2 per HPF; and BLM 6.4 ± 1.1 per HPF); however, a significantly increased number of apoptotic tumor cells was found in the Sham group (14.4 ± 2.7 per HPF) (Figure 4).

### 3.5. Tumor Cell Proliferation

The immunohistochemical analysis of tumor cell proliferation revealed up to 50% PCNA-positive cells in the rEP, BML, and Sham groups (rEP 3.6 ± 0.2; BLM 3.4 ± 0.3; and Sham 3.3 ± 0.3). Application of ECT lowered the number of proliferating tumor cells compared to the other groups, although the difference did not reach a significant level (ECT 2.0 ± 0.4) (Figure 5).

### 3.6. Tumor Vascularization

Tumor vascularization was examined with the identification of CD31-positive blood vessels. CD31 is expressed exclusively on the endothelial cells of blood vessels and can be utilized as an indicator of tissue vascularization. Treatment with ECT significantly reduced the tumor vascularization by 32–35% compared to the monotherapies with rEP or BLM (ECT 65.1 ± 12.1 mm^−2^; rEP 100.6 ± 7.7 mm^−2^; and BLM 96.5 ± 4.7 mm^−2^). Of interest, the treatment with rEP and BLM did not induce a reduction in the number of CD31-positive blood vessels compared to the Sham group (107.8 ± 15 mm^−2^) (Figure 6).

### 3.7. Inflammatory Response

The immunohistochemical detection of MPO-positive neutrophilic granulocytes revealed a decreased number of inflammatory cells in the tumor tissue following ECT treatment compared to the rEP and BLM group (ECT 7.80 ± 2.6 cells per HPF; rEP 14.9 ± 1.4 per HPF; and BLM 18.6 ± 1.9 per HPF). The analysis of inflammatory reaction further showed no difference between the rEP and BLM group; however, the greatest number of MPO-positive cells was detected in the Sham group, where the immune response was statistically increased compared to the other three groups (Sham 26.0 ± 1.9 per HPF) (Figure 7).

### 3.8. Blood Cell Analysis

The analysis of white blood cell count, including leukocytes, lymphocytes, monocytes, and neutrophils, revealed physiological values on day 13 in all groups. In the three treatment groups, leukocytes and lymphocytes were found increased compared to the Sham group, although the values did not exceed the normal ranges. RBC and platelet counts were within normal ranges in all groups. Finally, hemoglobin and hematocrit levels were also normal in the groups; however, slightly increased values were seen in the BLM group compared to the ECT, rEP, and Sham group (Table 1).

### 3.9. Adverse Events

Complications related to electrode insertion, electric pulse delivery, or BLM administration were not detected in our study. Animals showed no clinical signs of systemic inflammatory response or liver failure at any point during the study.

## 4. Discussion

Several ablative methods have been developed for the treatment of hepatic tumors, including irreversible EP, MWA, and RFA. In irreversible EP, membrane integrity is irreversible deranged through short pulses of high frequency direct current, inducing cell death [8]. In contrast to rEP, irreversible EP leads to an abnormal transmembrane electrical potential across the cellular membrane causing permanent loss of cellular homeostasis [25]. Clinical studies in the treatment of primary or metastatic hepatic tumors reported unfavourable results following irreversible EP. Specifically, local recurrence-free survival rate at one year was estimated at 59.5–70%, while increased incidence of recurrence was reported in tumors with diameter larger than 4 cm [26,27].

RFA and MWA, which are currently used in clinical practice, are prone to the heat-sink effect and are not indicated for the treatment of hepatic lesions adjacent to high-risk areas in the proximity of large vessels [1,4,8]. Furthermore, both procedures are associated with thermal-induced complications, which prevent their use for exophytic tumors as well as those in the vicinity of liver hilum, main bile ducts, gallbladder, or neighboring intestine [2,8]. Finally, decreased efficacy is reported in tumors larger than 30 mm, where tumor necrosis is estimated at 80% following treatment [6].

Therefore, ECT should be studied as a potentially ideal ablation technique for tumors in the vicinity of high-risk areas, since tumor necrosis is achieved through a nonthermal treatment modality, which preserves the patency of the vessels and bile ducts. Although retrospective studies with a limited number of patients have been conducted to evaluate the role of ECT in liver tumors [16], the mechanism of action of ECT on liver tissue based on immunohistochemical examinations and photoacoustic imagining remains unclear. In line with this view, we evaluated in the present study the effectiveness of ECT as a treatment approach for hepatic tumors.

Initially, we proved that the included animals presented similar tumor development and tumor characteristics. The sonographic and photoacoustic examination on the day of treatment revealed that the tumor volume, tumor oxygenation, and HbT were similar among the studied groups. Consequently, the different treatment approaches (ECT, rEP, and BLM) were examined in tumors with the same biology and comparable volume and vascularization. Furthermore, differences or significant reduction in body weight were not observed in the treated animals during the experiments.

Our results show that ECT is an effective treatment procedure with 87.2 ± 3.1% necrosis of the tumor tissue. The combination of BLM with rEP (as an ECT treatment modality) increased the effectiveness of BLM by ca. 355%. Similar results were reported in a meta-analysis with 44 studies and 1894 cutaneous and subcutaneous tumors, where the objective response rate to ECT was estimated at 84.1% compared to 19.9% for tumors treated only with chemotherapy [14].

Necrotic tissue was observed at the tumor tissue that had been exposed to the electrical pulses, whereas necrosis was not detected in the normal hepatic parenchyma, where the external electric field was not applied. This finding is attributed to the mechanism of action of rEP in combination with BLM. Firstly, the application of an external electric field through rEP induces a locally increased cellular permeability for hydrophilic agents, which are otherwise poorly penetrating chemotherapeutic agents, such as BLM and cisplatin. Secondly, BLM predominantly affects actively dividing cancer cells at the stage of mitosis (cell cycle G2) and, to a lesser extent, the normal non-dividing cells in the surrounding normal hepatic tissue [28]. Therefore, cytotoxic effects are observed in the tumor tissue, where the electric field is applied, without affecting the normal hepatic parenchyma. The negligible damage of normal hepatic tissue after ECT has been confirmed in other preclinical studies [29,30]. Specifically, necrosis was observed at the treated tumor tissue, whereas minimal histological changes with well-defined coagulation necrosis were detected around the inserted electrodes at the normal hepatic tissue.

The locally increased potentiation of BLM at the area of electric field application leads to an increased efficacy of the chemotherapy with reduced cumulative doses of the cytotoxic agent and limited side-effects, while the function of surrounding normal cells, tissues, and organs is preserved. BLM is associated with limited antitumor effectiveness when it is administrated as monotherapy. This is due to the reduced permeabilization through the cellular membrane [18]. However, if diffusion into the cytosol is achieved, BLM possesses a very potent intrinsic cytotoxicity. BLM induces cellular mitotic death through DNA single- and double-strand breaks with DNA fragmentation and chromosomal gaps.

In our study, the number of apoptotic cells was comparable among the three treatment groups. ECT with BLM did not lead to caspase-3 activation, revealing that ECT-mediated cellular death is not apoptosis-dependent. The mechanism of ECT-mediated cell death remains unclear and studies suggest that the cytotoxic effects of ECT are likely to be cell-specific [31,32]. ECT combined with BLM is not a treatment modality that can restore the apoptotic signaling pathways to eliminate the development of cancer cells.; therefore, the combination of ECT with anticancer treatments that target the apoptotic pathway, such as Bcl-2 family of proteins, p53, or IAPs, should be evaluated in further preclinical studies.

Apoptosis was detected reduced in the treatment groups compared to the Sham group. Apoptosis is an energy-demanding process, requiring intracellular adenosine triphosphate for the execution of cellular death [33]. Mitochondria are important organelles that play a key role in generating the energy required in the pathway of apoptosis [34]. It has been confirmed that application of an external electric field causes EP of intracellular membranes by altering the nuclear and mitochondrial transmembrane voltage [35,36]. In general, permeabilization of the cellular or mitochondrial membrane leads to activation of an energy-dependent process for membrane repair [36]. Our hypothesis is that this energy-repair process in the ECT and rEP groups reduces the energy reserves of the cells, leading to a limitation of other energy-demanding cellular functions such as apoptosis. In the BLM group, although drug penetration is limited in non-permeabilized cells, diffusion through the cell membrane will cause DNA single- and double-strand breaks [18]. We assume that this interaction of the malignant cells with BLM will cause an energy-dependent defence mechanism in the survived cells, which will reduce their ability to conduct apoptosis.

Photoacoustic imaging was used to provide a high-resolution 3D tomographic map of oxygen saturation by measuring oxygenated and deoxygenated hemoglobin. Tumor oxygenation decreased in the ECT group by 35.5% compared to the pretreatment values, which is attributed to the extensive necrosis of the tumor tissue as well as to the ECT vascular effects. Furthermore, ECT was the only treatment that caused a reduction in HbT of the tumor tissue by 7%, despite the fact that the difference in pre- and post-treatment values did not reach a significant level. In rEP, BLM, and Sham groups, HbT was increased by 53.6%, 53.9%, and 47.3%, respectively. This finding is attributed to the increased neovascularization of the tumor tissue that grew further within the five days following the ineffective treatments. However, although the concentration of oxygenated and deoxygenated hemoglobin was found increased in these groups, tumor oxygenation was irrespectively reduced compared to pretreatment values. This result is interpreted by the high oxygen consumption of the tumor cells that leads to lower oxygenation levels, despite the fact that the tumor vascularization and total HbT are increased.

On the other hand, the decreased HbT levels in the ECT group are attributed to the extensive tumor necrosis and to the effect of ECT on tumor perfusion. Tumor vascularization is affected by ECT through the “vascular disrupting effect” and the “vascular lock effect”. According to the “vascular disrupting effect”, the increased cell permeability through rEP is not limited to the tumor cells but extends to the endothelial vascular cells in the treated area. The subsequent increased diffusion of BLM into the endothelial cells leads to cellular damage, occlusion of blood flow, and ischemic death of the tumor cells adjacent to the obstructed blood vessel [37].

The “vascular lock effect” is characterized by vascular contraction. The electrical stimulation of precapillary smooth muscle cells causes direct vasoconstriction followed by indirect sympathetically mediated vasoconstriction of the afferent arterioles [9,11,37]. Furthermore, the shape of vascular endothelial cells is modified by the electrical fields, leading to increased vascular resistance and alteration of the endothelial cell-to-cell junctions [9]. As a result of the cell-to-cell junction disruption, protein leakage is observed, which leads to increased interstitial fluid pressure and decreased intravascular pressure. Finally, the reported hypoperfusion in the treated area has a beneficial effect on the treatment since it is not only associated with drug entrapment in the tumor tissue and an increased retaining time of the chemotherapeutic agent, but it also prevents bleeding during the intervention [9,11,37]. In agreement with the decrease of HbT on tumor tissue after ECT, CD31 immuno-staining showed that, in comparison with the other groups, tumor vascularization was reduced in the ECT group.

In addition, ultrasound imaging was utilized to identify specific sonographic findings as indicators of effective tumor treatment. Five days following the ECT treatment, tumors appeared sonographically as hypoechoic areas surrounded by a 3–5 mm hyperechoic rim. The hypoechoic zone was attributed to the necrotic tissue, the blockade of the tumor perfusion due to the electric pulse application, and to the water leakage out of the tissue, which caused tissue edema around the tumor and the normal hepatic tissue. The hepatic tissue around the electrodes also appeared as hypoechogenic areas due to the electrochemical reactions between electrodes and normal hepatic cells that induced cellular necrosis. Finally, the fibrotic zone at the peripheral parts of the necrotic tumor tissue presented as a 3–5 mm hyperechoic rim.

ECT has been associated with enhanced antitumor immunological response through release of intact tumor antigens from the damaged cells, inducing immunogenic cell death by tumor infiltration with immune cells [11,38]. ECT generates molecular patterns, including calreticulin membrane externalization and liberation of adenosine triphosphate and high mobility group box 1 protein, that trigger an immune response against the surviving malignant cells [38,39]. The immune cells that are involved in the local immune response are CD8^+^ T cells, CD4^+^ T cells, macrophages, and natural killer cells [38,39].

In our study, the inflammatory markers in blood cell analysis were found in normal ranges after treatment, which is in accordance with the results of a preclinical study in the liver tissue of pigs [29]. In the immunohistological examination, infiltration of neutrophilic granulocytes was decreased in the tumor tissue following ECT. We assume that this result is attributed to the increased necrosis of the malignant cells after ECT, which limits the need of granulocytes to proliferate and induce cellular death by phagocytosis. On the other hand, despite the fact that rEP and BLM groups presented decreased levels of granulocytes compared to the Sham group, there was a trend of increased infiltration of granulocytes also in those groups. In accordance to our results, preclinical and clinical studies showed limited infiltration of neutrophilic granulocytes after ECT, reporting mainly infiltration of lymphocytes and plasma cells at the margin of the ablated area and normal liver parenchyma [40,41].

In our study, ECT was a safe and well tolerated procedure without adverse events during the postinterventional period. The health status and the weight of the animals were not affected. Furthermore, despite the fact that the needles were inserted in the hepatic parenchyma, bleeding complications were not observed during or after treatment. This finding is in accordance with the blood cell analysis, where hemoglobin, hematocrit, and RBC count in the ECT group were similar to the other groups. The absence of bleeding is attributed to the transient local hypoperfusion as well as to the electrocoagulation at the needle insertion point, as a result of the high intensity current at the surface of the needles.

The utilization of ECT presents some limitations and restrictions. The most crucial factor for the effectiveness of ECT is the sufficient distribution of the electric field in the tumor tissue. The produced electric field is influenced by the electrical properties of the target tissue, the position and geometry of the electrodes, and the parameters of the electric pulses. Individual treatments protocols are required in order to achieve sufficient electric field distribution as well as increased concentration of the chemotherapeutic agent at the tumor tissue. In the case of large metastases, correct needle placement in order to ensure an effective and homogeneous coverage with the electric field is mandatory. Imaging modalities and software have been developed to optimize needle positioning and to predict the homogenous electric field distribution [11,42,43].

A second crucial factor for a successful ECT treatment is the intratumoral concentration of the chemotherapeutic agent. Large tumors and tissues with decreased or heterogeneous perfusion may present inhomogeneous concentrations of the chemotherapeutic agent and insufficient treatment. This restriction can be overcome with the simultaneous intratumoral and intravenous injection of the chemotherapeutic agent. In our next study, we will evaluate the combined administration in order to increase the effectiveness of the method.

The present study also exhibits several limitations. Following the 3R principle (replacement, reduction, and refinement), we performed our experiments only in a limited number of animals, which can affect the results and precluded more statistical analysis of the obtained data; moreover, our results are limited to the treatment of colorectal metastases. Further studies are required to assess the effectiveness of ECT in the treatment of tumors with other pathology. Specifically, hepatocellular cancer presents enhanced vascularization compared to colorectal metastases, which are more fibrotic [44]. This difference in tumor vascularization could lead to increased diffusion of BLM into tumor cells of hepatocellular cancer and to enhanced effectiveness of the treatment. Finally, although ECT is considered as a nonthermal ablation procedure, the thermal effect of the applied electric field in the liver tissue was not examined in this study.

## 5. Conclusions

ECT is a safe, feasible, and effective treatment procedure for hepatic tumors; hence, this ablation procedure may be applied in combination with parenchymal sparing hepatectomies, allowing radical treatment of primary unresectable tumors. Further studies are required to evaluate the role of ECT in the treatment of tumors in vicinity of major hepatic vascular structures, bile ducts, or viscera as well as for tumors with a pathology other than colorectal metastases. In addition, other chemotherapeutic agents than BLM indicated in the treatment of liver cancer should be studied in combination with ECT. 

## Figures and Tables

**Figure 1 cancers-15-01598-f001:**
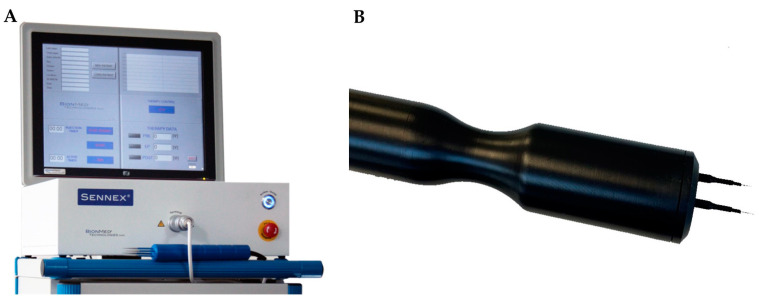
(**A**) Sennex^®^ Tumor System, BIONMED^®^ Technologies GmbH, Saarbrücken, Germany. (**B**) Fixed geometry linear needle electrodes in a plastic holder. (Images are utilized with consent and permission of BIONMED^®^ Technologies).

**Figure 2 cancers-15-01598-f002:**
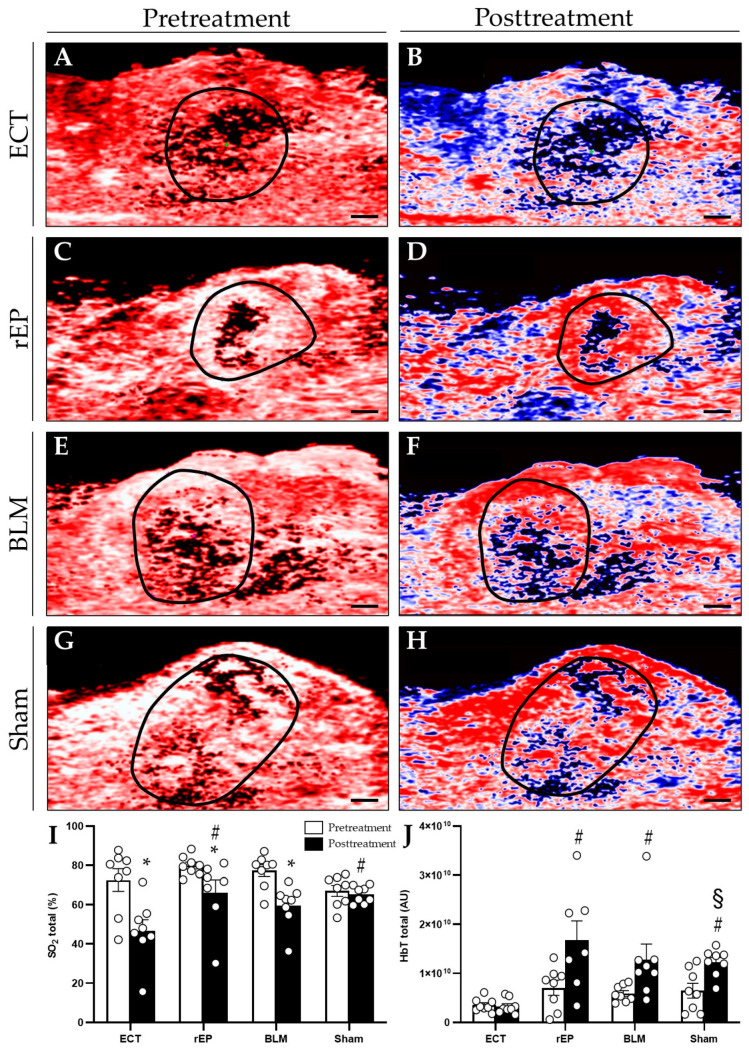
(**A**–**H**) Photoacoustic imaging. Pretreatment (on day 8) and posttreatment (on day 13) hemoglobin map in animals treated with ECT (**A**,**B**), rEP (**C**,**D**), BLM (**E**,**F**), or Sham (**G**,**H**). Scale bars: 1 mm. (**I**,**J**) SO_2_ and HbT were measured in the whole tumor area. Data are given as mean ± SEM; # *p* < 0.05 vs. ECT posttreatment; * *p* < 0.05 vs. pretreatment values in each group; § *p* < 0.05 vs. Sham pretreatment.

**Figure 3 cancers-15-01598-f003:**
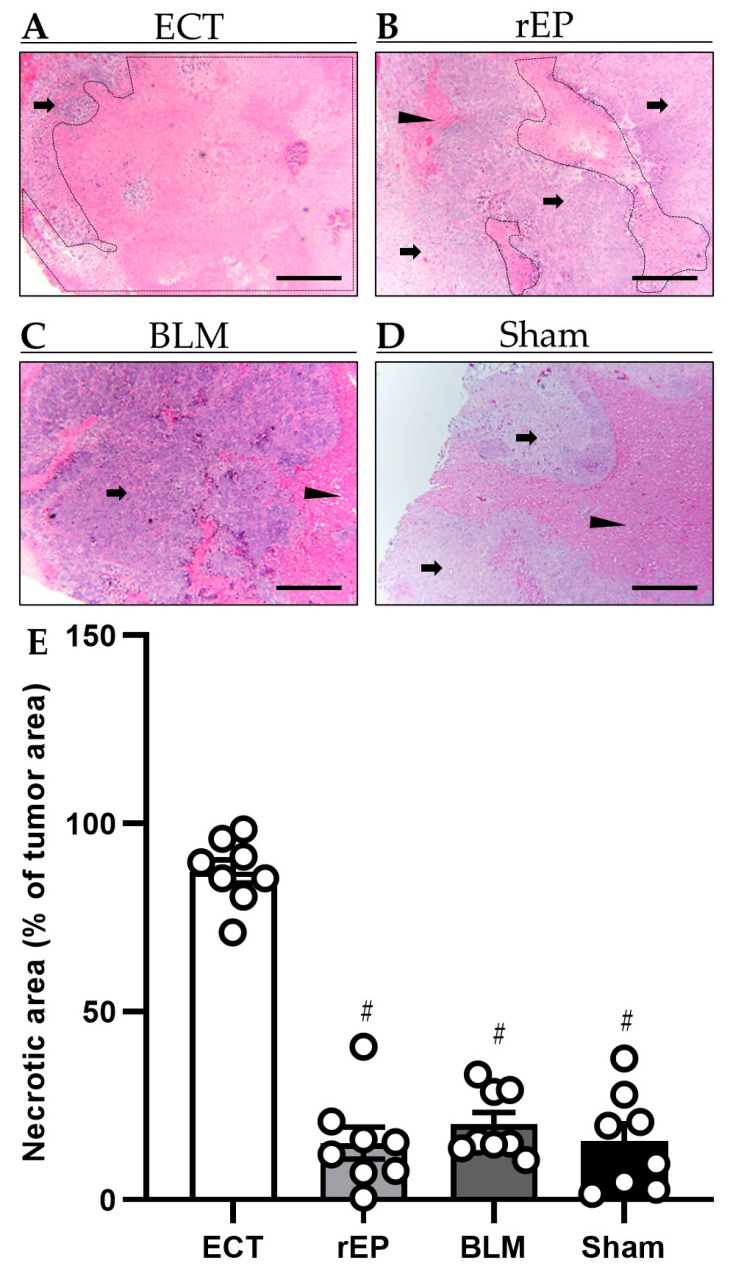
(**A**–**D**) Histological analysis of necrotic cell death (borders marked by dotted line) in animals treated with ECT (**A**), rEP (**B**), BLM (**C**), or Sham (**D**). Tumor tissue is marked by arrows and normal hepatic tissue by arrowheads. Scale bars: 500 µm. (**E**) Necrotic areas in the tumor tissue were measured as percentage of the whole tumor area. Data are given as mean ± SEM; # *p* < 0.05 vs. ECT.

**Figure 4 cancers-15-01598-f004:**
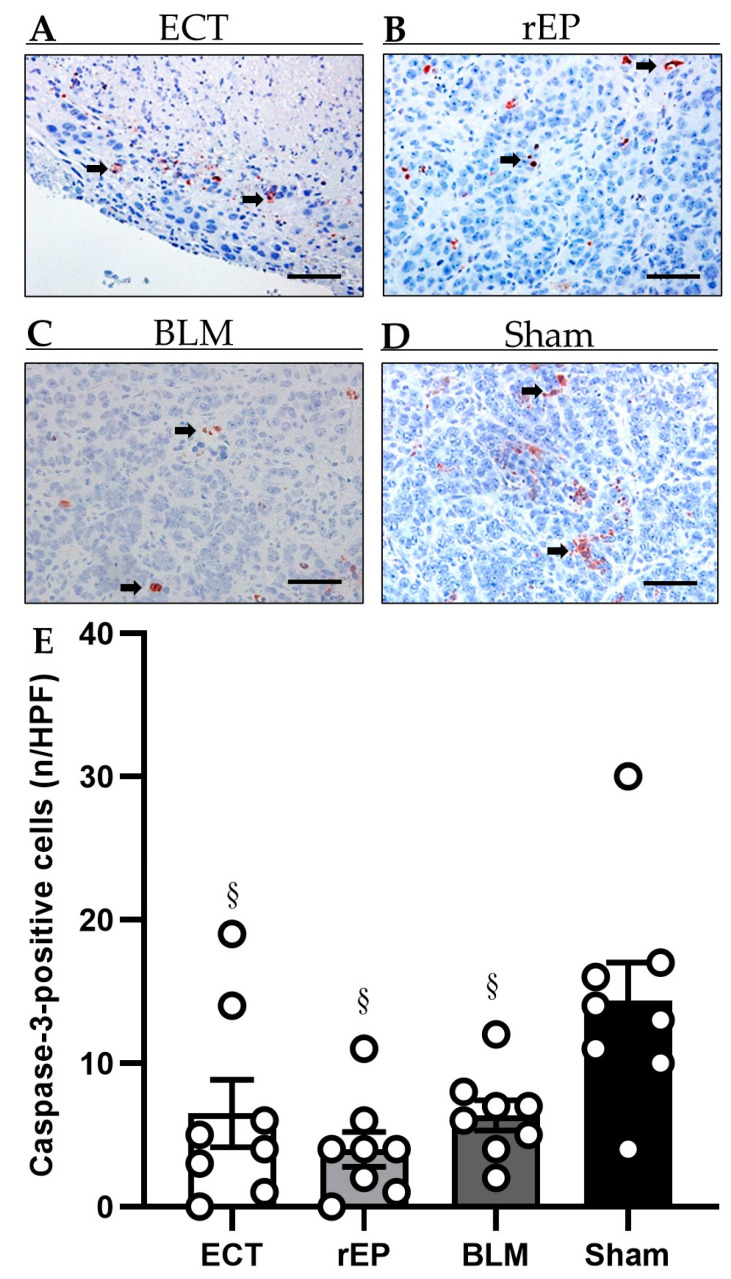
(**A**–**D**) Immunohistochemical analysis of cleaved caspase-3 in animals treated with ECT (**A**), rEP (**B**), BLM (**C**), or Sham (**D**). Apoptotic cells (arrows) are stained red. Scale bars: 50 µm. (**E**) The diagram displays the mean number of positive cells in the tumor tissue per HPF. Data are given as mean ± SEM; § *p* < 0.05 vs. Sham.

**Figure 5 cancers-15-01598-f005:**
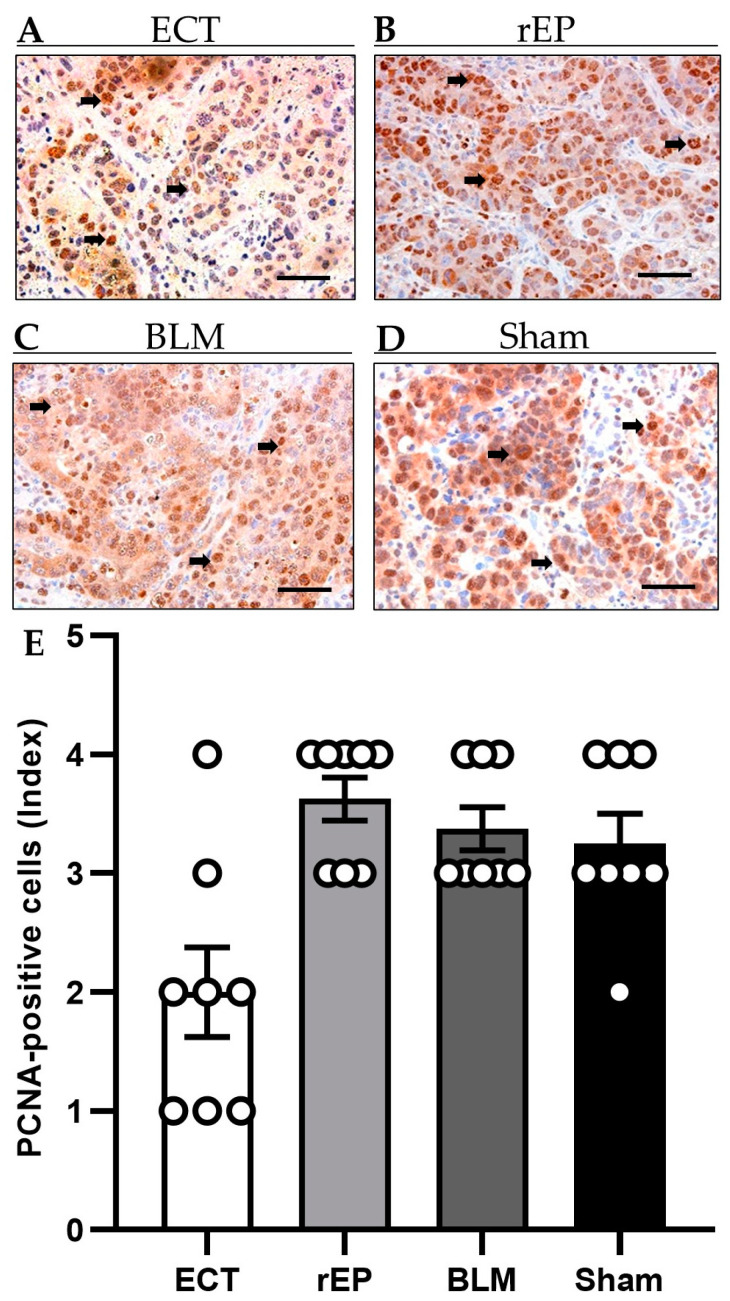
(**A**–**D**) Immunohistochemical sections of PCNA expression in the tumor tissue of animals undergoing ECT (**A**), rEP (**B**), BLM (**C**), or Sham (**D**). PCNA-positive cells (arrows) are stained brown. Scale bars: 50 µm. (**E**) The diagram displays the quantitative analysis of PCNA-positive cells (index) in the tumor tissue. Data are given as mean ± SEM.

**Figure 6 cancers-15-01598-f006:**
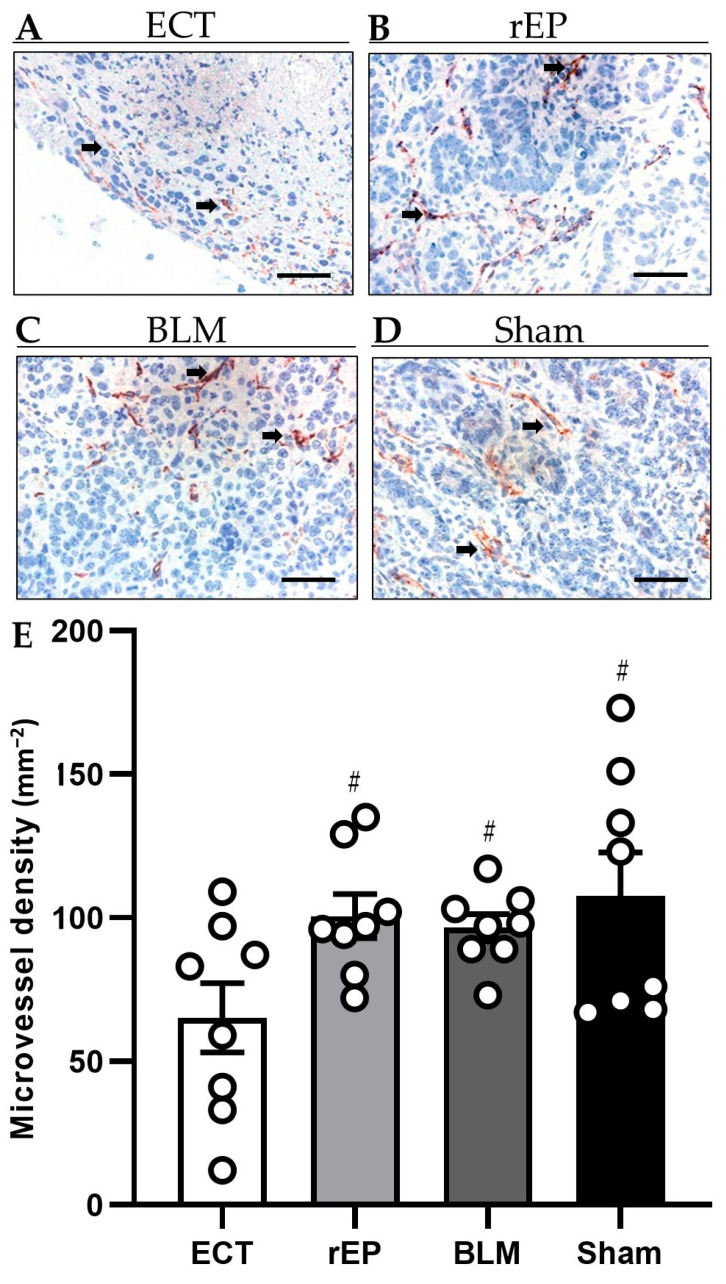
(**A**–**D**) Immunohistochemical analysis of CD31 expression in the tumor tissue of animals treated with ECT (**A**), rEP (**B**), BLM (**C**), and Sham (**D**). CD31-positive blood vessels (arrows) are stained red. Scale bars: 50 µm. (**E**) The diagram shows the quantitative analysis of CD31-positive blood vessels in tumor tissues. The microvessel density is given as the mean number of positive vessels (in 10 HPF) in mm^−2^; # *p* < 0.05 vs. ECT.

**Figure 7 cancers-15-01598-f007:**
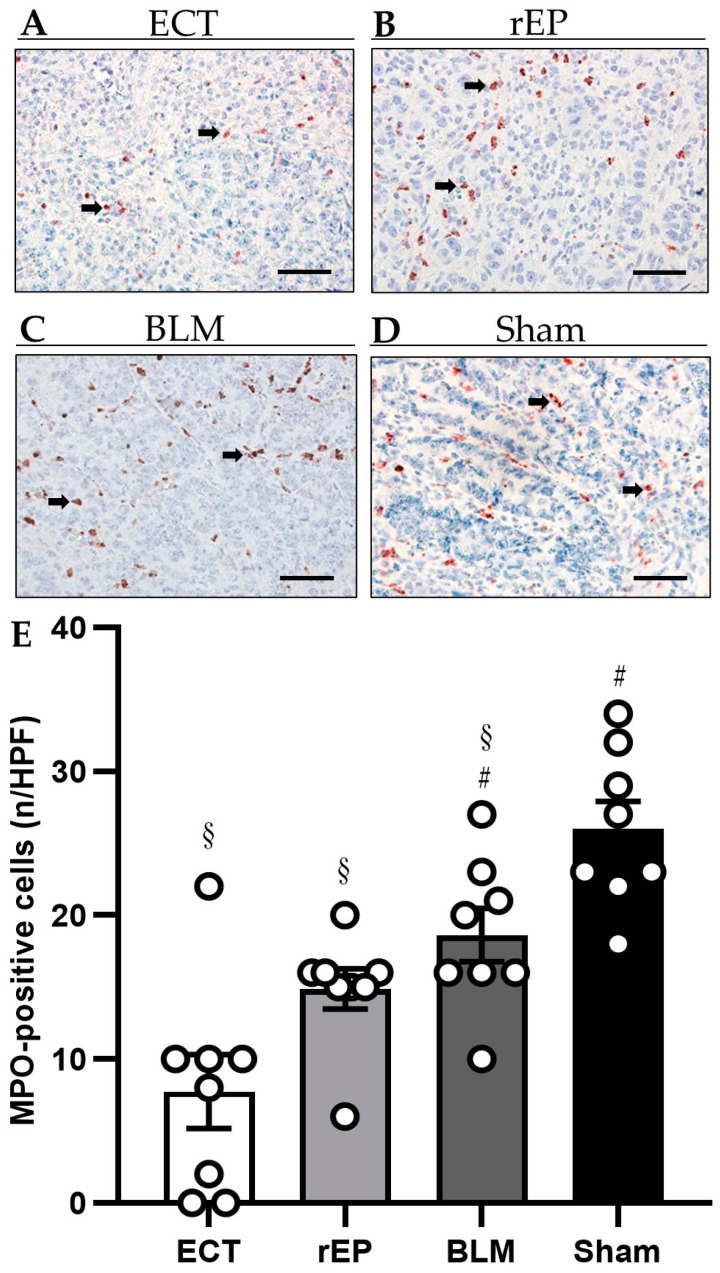
(**A**–**D**) Immunohistochemical analysis of MPO expression in the tumor tissue of animals treated with ECT (**A**), rEP (**B**), BLM (**C**), and Sham (**D**). MPO-positive cells are stained red (arrows). Scale bars: 50 µm. (**E**) The diagram displays the quantitative analysis of MPO-positive cells in tumor tissues. Data are given as mean ± SEM; # *p* < 0.05 vs. ECT; § *p* < 0.05 vs. Sham.

**Table 1 cancers-15-01598-t001:** Blood cell analysis on day 13.

	ECT	rEP	BLM	Sham
Leukocytes (10^9^/L)	8.9 ± 1 ^§^	7.8 ± 0.5 ^§^	8.7 ± 0.9 ^§^	4 ± 0.4
Lymphocytes (10^9^/L)	4.7 ± 0.2 ^§^	5 ± 0.4 ^§^	4.8 ± 0.5 ^§^	2.4 ± 0.3
Monocytes (10^9^/L)	0.8 ± 0.2	0.4 ± 0.1 ^#^	0.6 ± 0.1	0.1 ± 0.03 ^#^
Neutrophils (10^9^/L)	3.4 ± 0.8	2.4 ± 0.3	3.3 ± 0.6	1.6 ± 0.3 ^#^
RBC (10^12^/L)	6.8 ± 0.2	7.1 ± 0.4	7.9 ± 0.5 ^#^	6.9 ± 0.3
Platelets (g/dL)	606.5 ± 24.7	626.6 ± 35.8	551.6 ± 29.5	514.7 ± 74.2
Hemoglobin (g/dL)	13.2 ± 0.4 ^▲^	13.3 ± 0.6 ^▲^	15.7 ± 0.8	11.7 ± 0.4 ^▲^
Hematocrit (%)	38.4 ± 0.9 ^▲^	38.2 ± 1.6 ^▲^	42.9 ± 2.2	36.3 ± 0.9 ^▲^

Data are given as mean ± SEM; ^#^ *p* < 0.05 vs. ECT; ^§^ *p* < 0.05 vs. Sham; ^▲^ vs. BLM.

## Data Availability

The data presented in the study are available from the corresponding author upon reasonable request.

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
