# Peer review of "Evaluation of Electrochemotherapy with Bleomycin in the Treatment of Colorectal Hepatic Metastases in a Rat Model"

_cancers, 2023, doi:10.3390/cancers15051598_

Round 1
Reviewer 1 Report
Interesting work by authors. I have a few questions:
1. Authors used rats from both gender? How were both genders divided into four groups? Were there similar number of males/females in four groups?
2. How many animals per group were used? It is not clear to me.
3. Not sure how applying a 1000V across 8mm inter-electrode distance will create a 1000 V/cm E-field. It will generate 1250V/cm. "The voltage between the pair of electrodes was set to 136 1.000 V, corresponding to an amplitude of 100 V/mm and frequency of 1Hz". Please clarify.
4. Authors highlight thermal safe nature of ECT. How do you know this type of electrodes used in this study is thermally safe? In addition of the voltage, pulse width and other electrical pulse parameters, electrodes dimension and geometry also play a critical role in thermal effect. There is always some thermal effect when electrical pulses are applied. Since authors do not include such thermal effect data, I am not sure how this method described with a particular geometry is thermal free. Could Authors please clarify? If not possible to include thermal data, it could be discussed in the limitation of the study.
4. Not sure how by just monitoring for one Apoptotic marker, authors could conclude about the absence of apoptosis for ECT. As I understand Apoptosis could be a complex phenomenon with often a lot of interactions with other cell death pathways (such as necrosis, apoptosis, necroptosis, ferroptosis), so not sure by only studying one proteins, authors could conclude the presence/absence of apoptosis - in fact there is a whole research field dedicated to study these. Could authors clarify?
Author Response
Review of the manuscript ID ‘cancers-2184116’ by Spiliotis et al.
Reply to the comments of reviewer 1
Reviewer comment 1: Authors used rats from both gender? How were both genders divided into four groups? Were there similar number of males/females in four groups?
Reply: In our study, rats from both gender were utilized for the experiments (males n = 16, females n = 16). Animals of both gender were equally distributed in each group (males n = 4, females n = 4). This information is provided in the revised form of the paper.
(See page 2 and 3, line 89 and 101)
Reviewer comment 2: How many animals per group were used? It is not clear to me.
Reply: Eight animals were utilized in each group. Thirty-two rats were utilized in total. This information was provided in the submitted version of the paper in the section of Material and Methods.
(See page 2, line 89)
Reviewer comment 3: Not sure how applying a 1000V across 8mm inter-electrode distance will create a 1000 V/cm E-field. It will generate 1250V/cm. "The voltage between the pair of electrodes was set to 136 1.000 V, corresponding to an amplitude of 100 V/mm and frequency of 1Hz". Please clarify.
Reply: The reviewer is correct that the amplitude is estimated at 1250 V/cm. We contacted the company BIONMED®Technologies GmbH, which produces the ECT device (Sennex® Tumor-System). They confirmed that the voltage between the pair of electrodes was set to 1.000 V, corresponding to an amplitude of 1250 V/cm (125 V/mm) and frequency of 1 Hz. The amplitude has been corrected in the revised form of the paper.
(See page 3 and 4, line 147-149)
Reviewer comment 4: Authors highlight thermal safe nature of ECT. How do you know this type of electrodes used in this study is thermally safe? In addition of the voltage, pulse width and other electrical pulse parameters, electrodes dimension and geometry also play a critical role in thermal effect. There is always some thermal effect when electrical pulses are applied. Since authors do not include such thermal effect data, I am not sure how this method described with a particular geometry is thermal free. Could Authors please clarify? If not possible to include thermal data, it could be discussed in the limitation of the study.
Reply: According to the literature, ECT is a thermal safe procedure, where pore formation and membrane destabilization are transient, and cells regain homeostasis (Miklavčič et al., 2014; Geboers et al., 2020). The reviewer is correct and rEP can induce irreversible thermal injury. The time and the applied voltages on the surface of the tumors have significant effects on the absolute temperature and the quantity of the irreversible thermal damage (Youssef et al., 2020). However, in preclinical studies, no significant tissue heating has been observed in liver treated with ECT, where the highest reported temperature had been estimated at 47 °C (Cindric et al., 2022).
As reported in our results, an ablated area was detected at the electrodes’ insertion site, which was oval with a diameter of 3-4 mm. In these foci, the hepatic tissue was necrotic, including perforated blood vessels or dilated capillaries with RBC extravasates, and was surrounded by inflammatory cells. This finding could be attributed to irreversible thermal injury or irreversible electroporation of the liver tissue adjacent to the electrodes.
We did not perform analyses to evaluate the thermal effect of the applied electric field on the liver tissue, since that was not within the scope of our study. For that reason, according to the comment of the reviewer, we included this information as limitation of our study in the Discussion section of the revised manuscript, which reads as follows:
‘Finally, although ECT is considered as a nonthermal ablation procedure, the thermal effect of the applied electric field in the liver tissue was not examined in this study.’
(See page 16, line 550-552)
References:
Miklavčič D, Mali B, Kos B, Heller R, Serša G. Electrochemotherapy: from the drawing board into medical practice. Biomed Eng Online. 2014;13(1):29.
Geboers B, Scheffer HJ, Graybill PM, Ruarus AH, Nieuwenhuizen S, Puijk RS, et al. High-Voltage Electrical Pulses in Oncology: Irreversible Electroporation, Electrochemotherapy, Gene Electrotransfer, Electrofusion, and Electroimmunotherapy. Radiology. 2020;295(2):254-72.
Youssef HM, El-Bary AA. Voltage and Time Required for Irreversible Thermal Damage of Tumor Tissues during Electrochemotherapy under Thomson Effect. Mathematics. 2020; 8(9):1488.
Cindric H, Gasljevic G, Edhemovic I, Brecelj E, Zmuc J, Cemazar M, et al. Numerical mesoscale tissue model of electrochemotherapy in liver based on histological findings. Sci Rep. 2022;12(1):6476.
Reviewer comment 5: Not sure how by just monitoring for one Apoptotic marker, authors could conclude about the absence of apoptosis for ECT. As I understand Apoptosis could be a complex phenomenon with often a lot of interactions with other cell death pathways (such as necrosis, apoptosis, necroptosis, ferroptosis), so not sure by only studying one proteins, authors could conclude the presence/absence of apoptosis - in fact there is a whole research field dedicated to study these. Could authors clarify?
Reply: The characteristic features that define apoptosis are dependency of caspase activation and cleavage of specific cellular proteins or “death” substrates within the cell. Apoptosis may can be viewed, in biochemical terms, as a caspase-mediated form of cell death (Chakraborty et al., 2012).
Apoptosis includes two major pathways: (a) intrinsic or mitochondrial and (b) extrinsic or death receptor related. Cleaved caspase-3, which has been utilized in our study, is a biochemical marker that is activated in both intrinsic and extrinsic pathway of apoptosis. Although apoptosis is a complex cellular procedure, cleaved caspase-3 represents a crucial apoptotic marker that can be used as an indicator of the cellular apoptosis in liver tissue (Hussar, 2022). Furthermore, cleaved caspase-3 immunostaining can identify hepatocytes at a very early apoptotic stage, where they show cytoplasmic staining and an otherwise intact cellular structure (Eckle et al., 2004).
In the following figure published by Wu et al., it is demonstrated that caspase-3 is a crucial factor in the final steps of both apoptotic cascades. Caspase-3 is directly activated through the cleaved caspase-8 in the extrinsic pathway. In the intrinsic pathway, caspase-3 is activated both directly by caspase-9 and indirectly by caspase-8 in a mitochondrion-dependent manner via caspase-9 (Wu et al., 2016).
Furthermore, several studies have been conducted to evaluate the role of different treatments in hepatic diseases in rat models. The majority of those studies had utilized cleaved caspase-3 for the immunohistochemical detection of apoptotic cells (Sperling et al., 2013; Sperling et al., 2015; Ziemann et al., 2019; Kauffels et al., 2019; von Heesen et al., 2015; Somade et al., 2020; Zhu et al., 2019; Esmat et al., 2022). Based on the current literature and in order to eventually compare our results with the results of other studies, which has utilized the same apoptotic marker, we used cleaved caspase-3 in our experiments.
References:
Chakraborty JB, Oakley F, Walsh MJ. Mechanisms and biomarkers of apoptosis in liver disease and fibrosis. Int J Hepatol. 2012;2012:648915.
Hussar P. Apoptosis Regulators Bcl-2 and Caspase-3. Encyclopedia. 2022; 2(4):1624-1636.
Eckle VS, Buchmann A, Bursch W, Schulte-Hermann R, Schwarz M. Immunohistochemical detection of activated caspases in apoptotic hepatocytes in rat liver. Toxicol Pathol. 2004;32(1):9-15.
Wu Y, Zhao D, Zhuang J, Zhang F, Xu C. Caspase-8 and Caspase-9 Functioned Differently at Different Stages of the Cyclic Stretch-Induced Apoptosis in Human Periodontal Ligament Cells. PLoS One. 2016;11(12):e0168268.
Sperling J, Brandhorst D, Schafer T, Ziemann C, Benz-Weisser A, Scheuer C, et al. Liver-directed chemotherapy of cetuximab and bevacizumab in combination with oxaliplatin is more effective to inhibit tumor growth of CC531 colorectal rat liver metastases than systemic chemotherapy. Clin Exp Metastasis. 2013;30(4):447-55.
Sperling J, Ziemann C, Gittler A, Benz-Weisser A, Menger MD, Kollmar O. Tumour growth of colorectal rat liver metastases is inhibited by hepatic arterial infusion of the mTOR-inhibitor temsirolimus after portal branch ligation. Clin Exp Metastasis. 2015;32(4):313-21.
Ziemann C, Roller J, Malter MM, Keller K, Kollmar O, Glanemann M, et al. Intra-arterial EmboCept S(R) and DC Bead(R) effectively inhibit tumor growth of colorectal rat liver metastases. BMC Cancer. 2019;19(1):938.
Kauffels A, Kitzmuller M, Gruber A, Nowack H, Bohnenberger H, Spitzner M, et al. Hepatic arterial infusion of irinotecan and EmboCept® S results in high tumor concentration of SN-38 in a rat model of colorectal liver metastases. Clin Exp Metastasis. 2019;36(1):57-66.
von Heesen M, Dold S, Muller S, Scheuer C, Kollmar O, Schilling MK, et al. Cilostazol improves hepatic blood perfusion, microcirculation, and liver regeneration after major hepatectomy in rats. Liver Transpl. 2015;21(6):792-800.
Somade OT, Ajayi BO, Olunaike OE, Jimoh LA. Hepatic oxidative stress, up-regulation of pro-inflammatory cytokines, apoptotic and oncogenic markers following 2-methoxyethanol administrations in rats. Biochem Biophys Rep. 2020;24:100806.
Zhu Y, Xiang T, Hu D, Shi B, Zhang W, Zhang S, et al. Protective effects of mild hypothermia against hepatic injury in rats with acute liver failure. Ann Hepatol. 2019;18(5):770-6.
Esmat MA, Osman A, Hassan RE, Hagag SA, El-Maghraby TK. Hepatoprotective effect of ferulic acid and/or low doses of gamma-irradiation against cisplatin-induced liver injury in rats. Hum Exp Toxicol. 2022;41:9603271221136205.

Author Response
Review of the manuscript ID ‘cancers-2184116’ by Spiliotis et al.
Reply to the comments of reviewer 2
We appreciate the fair and constructive comments of the reviewer. In the following, please find our point-by-point reply.
Major Revision
Reviewer comment 1: Need to comment on the disadvantages of ECT (i.e., drug distribution, drug resistance, narrow reversible zone) in order to compare this method to other ablation modalities.
Reply: According to the comment of the reviewer, we have included an overview of the disadvantages and limitations of ECT in the Discussion section of our revised manuscript, which reads as follows:
‘The utilization of ECT presents some limitations and restrictions. The most crucial factor for the effectiveness of ECT is the sufficient distribution of the electric field in the tumor tissue. The produced electric field is influenced by the electrical properties of the target tissue, the position and geometry of the electrodes, and the parameters of the electric pulses. Individual treatments protocols are required in order to achieve sufficient electric field distribution as well as increased concentration of the chemotherapeutic agent at the tumor tissue. In the case of large metastases, correct needle placement to ensure an effective and homogeneous coverage of the electric field is mandatory. Imaging modalities and software have been developed to optimize needle positioning and to predict the homogenous electric field distribution (Campana et al., 2019; Probst et al., 2018; Marcan et al., 2015).
A second crucial factor for a successful ECT treatment is the intratumoral concentration of the chemotherapeutic agent. Large tumors and tissues with decreased or heterogeneous perfusion may present inhomogeneous concentrations of the chemotherapeutic agent and insufficient treatment. This restriction can be overcome with the simultaneous intratumoral and intravenous injection of the chemotherapeutic agent. In our next study, we will evaluate the combined administration in order to increase the effectiveness of the method.’
(See page 15, line 525-541)
References:
Campana LG, Edhemovic I, Soden D, Perrone AM, Scarpa M, Campanacci L, et al. Electrochemotherapy - Emerging applications technical advances, new indications, combined approaches, and multi-institutional collaboration. Eur J Surg Oncol. 2019;45(2):92-102.
Probst U, Fuhrmann I, Beyer L, Wiggermann P. Electrochemotherapy as a New Modality in Interventional Oncology: A Review. Technol Cancer Res Treat. 2018;17:1533033818785329.
Marcan M, Kos B, Miklavcic D. Effect of blood vessel segmentation on the outcome of electroporation-based treatments of liver tumors. PLoS One. 2015;10(5):e0125591.
Reviewer comment 2: It is a big claim to say that this study is testing ECT as a new strategy for hepatic tumors, when there is other literature testing this in a similar fashion. Suggestion to remove ‘new’ and more suggest how this paper will help continue to give insight into hepatic colorectal metastasis. Likewise (Line 71): Clinical studies of ECT in the liver have already been established. (https://www.ncbi.nlm.nih.gov/pmc/articles/PMC8884851/; https://biomedical-engineering-online.biomedcentral.com/articles/10.1186/1475-925X-14-S3-S5; https://www.mdpi.com/2075-4418/13/2/209;)
Reply: We agree with the reviewer that ECT has been utilized in the treatment of hepatic tumors in preclinical and clinical studies. However, the immunohistological and photoacoustic examinations following ECT in hepatic tumors, which have been conducted in our study, had not been reported in previous studies.
According to the comment of the reviewer, we have rephrased our statement in the Introduction and Conclusion section, which now reads as follows:
‘Retrospective studies with limited number of patients have been conducted to evaluate the role of ECT in liver tumors (Spallek et al., 2022). However, immunohistological examinations and photoacoustic imaging following ECT in liver tissue are poorly investigated as well as the role of the procedure in colorectal metastases.’
(See page 2, line 80-84)
‘Although retrospective studies with limited number of patients have been conducted to evaluate the role of ECT in liver tumors (Spallek et al., 2022), the mechanism of action of ECT on liver tissue based on immunohistochemical examinations and photoacoustic imagining remains unclear. In line with this view, we evaluated in the present study the effectiveness of ECT as a treatment approach for hepatic tumors.
(See page 13, line 387-392)
Reference:
Spallek H., Bischoff P., Zhou W., de Terlizzi F., Jakob, F., Kovacs A. Percutaneous electrochemotherapy in primary and secondary liver malignancies - local tumor control and impact on overall survival. Radiol Oncol 2022, 56 (1), 102-110.
Reviewer comment 3: Some of the presented results contradict previously published literature.
- It appears that the immune response is suppressed by ECT and bleomycin. Previous studies suggest that ECT with bleomycin improve immune activation (https://doi.org/10.4161/onci.28131; please see section 3.4 of this review for more references https://www.mdpi.com/2072-6694/14/12/2876). Please comment on this in your discussion.
- Figure 2: How were pre- and post-treatment significant differences only notable in the sham group?
- Figure 4: Why would there be an increased number of apoptotic cells in the sham group?
Reply:
- a) We agree with the reviewer that ECT improves the immune response through release of intact tumor antigens from the damaged cells. However, the immune cells that are involved in the local immune response are CD8+ T cells, CD4+ T cells, macrophages, and natural killer cells (Calvet et al., 2014; Justesen et al., 2022). We did not perform immunohistological analyses to evaluate the infiltration of different immune cells, since this was not within the scope this study. Nonetheless, based on the finding of this study, we know that neutrophilic granulocytes are decreased in the tumor tissue, which is maybe attributed to the increased necrosis following ECT and the limited phagocytosis of malignant cells by granulocytes.
According to the comment of the reviewer, we have included a novel paragraph in the Discussion section of our revised manuscript, which reads as follows:
‘ECT has been associated with enhanced antitumor immunological response through release of intact tumor antigens from the damaged cells, inducing immunogenic cell death by tumor infiltration with immune cells (Justesen et al., 2022; Probst et al., 2018). ECT generates molecular patterns, including calreticulin membrane externalization and liberation of adenosine triphosphate and high mobility group box 1 protein, that trigger an immune response against the surviving malignant cells (Calvet et al., 2014; Justesen et al., 2022). The immune cells that are involved in the local immune response are CD8+ T cells, CD4+ T cells, macrophages, and natural killer cells (Calvet et al., 2014; Justesen et al., 2022).
In our study, the inflammatory markers in blood cell analysis were found in normal ranges after treatment, which is in accordance with the results of a preclinical study in the liver tissue of pigs (Zmuc et al., 2019). In the immunohistological examination, infiltration of neutrophilic granulocytes was decreased in the tumor tissue following ECT. We assume that this result is attributed to the increased necrosis of the malignant cells after ECT, which limits the need of granulocytes to proliferate and induce cellular death by phagocytosis. On the other hand, despite the fact that rEP and BLM groups presented decreased levels of granulocytes compared to the Sham group, there was a trend of increased infiltration of granulocytes also in those groups. In accordance to our results, preclinical and clinical studies showed limited infiltration of neutrophilic granulocytes, reporting mainly infiltration of lymphocytes and plasma cells at the margin of the ablated area and normal liver parenchyma (Gasljevic et al., 2017, Chazal et al., 1998).’
(Page 15 and 16, lines 498-516)
- b) We thank the reviewer for this comment. The increase in hemoglobin levels compared to pretreatment values was calculated incorrectly in the rEP and BLM groups. Specifically, the hemoglobin levels in the rEP, BLM and Sham groups were increased by 53.6%, 53.9%, and 47.3% compared to pretreatment values in each group, respectively.
The difference was statistically significant only in the Sham group. Despite the fact that the 3 groups presented approximately the same increase in hemoglobin levels, the values in the rEP and BLM groups presented increased standard deviation leading to no statistically significant results. For that reason, even though the 3 groups presented similar tendency in posttreatment hemoglobin levels, only in the Sham group we found statistically significant results.
The correct percentages are provided in the revised form of the manuscript.
(See page 6, line 264-266)
- c) In our study, the number of apoptotic cells was similar among the three treatment groups and reduced compared to the Sham group. ECT with BLM did not lead to caspase-3 activation, revealing that ECT-mediated cellular death is not apoptosis-dependent. The mechanism of ECT-mediated cell death remains unclear and studies suggest that the cytotoxic effects of ECT are likely to be cell-specific (Fernandes et al., 2019; Batista Napotnik, 2021). Evidence regarding cellular death following ECT in liver cancer in limited. Two clinical studies with a total of 14 hepatic colorectal metastases showed necrosis of the treated tumors without reported apoptosis (Edhemovic et al., 2011; Edhemovic et al., 2014). According to our knowledge, no preclinical studies in animal models examined with immunohistochemical examinations the effects of ECT in hepatic tumors.
In an in-vitro study, ECT in combination with BLM, oxaliplatin, or cisplatin was examined in human and murine pancreatic cancer cell lines (Fernandes et al., 2019). Based on morphological and biochemical alterations, cancer cells did not appear to undergo apoptosis but instead necroptosis. Necroptosis is a novel form of regulated cell death not mediated by caspases, which requires the activity of receptor-interacting serine/threonine-protein kinases (de Almagro et al., 2015). Similar to our results, caspase-3 presented the highest activity in the control untreated group in both cancer lines.
Apoptosis is an energy-demanding process, requiring intracellular ATP for the execution of cellular death (Imamura et al., 2020). Mitochondria are important organelles that play key role in generating the energy required in the pathway of apoptosis (Zhou et al., 2022). It has been confirmed that application of external electric field causes nonthermal intracellular effects with EP of intracellular membranes, by altering the nuclear or mitochondrial transmembrane voltage (Esser et al., 2010, Zhou et al., 2021). In general, permeabilization of the cellular or mitochondrial membrane leads to activation of an energy-dependent process for membrane repair (Zhou et al., 2021). Our hypothesis is that this energy-repair process in the ECT and rEP groups reduces the energy reserves of the cells leading to a limitation of other energy-demanding cellular functions such as apoptosis.
In the BLM group, although drug penetration is limited in non-permeabilized cells, diffusion through the cell membrane will cause a BLM induced DNA single- and double-strand breaks (Bonferoni et al., 2021). We assume that this interaction of malignant cells with BLM will cause an energy-dependent defence mechanism in the survived cells, which will reduce their ability to conduct apoptosis.
This information is provided in the revised version of the manuscript.
(See page 13 and 14, lines 429-452)
References:
Fernandes P, O'Donovan TR, McKenna SL, Forde PF. Electrochemotherapy Causes Caspase-Independent Necrotic-Like Death in Pancreatic Cancer Cells. Cancers (Basel). 2019;11(8).
Batista Napotnik T, Polajzer T, Miklavcic D. Cell death due to electroporation - A review. Bioelectrochemistry. 2021;141:107871.
Edhemovic I, Gadzijev EM, Brecelj E, Miklavcic D, Kos B, Zupanic A, et al. Electrochemotherapy: a new technological approach in treatment of metastases in the liver. Technol Cancer Res Treat. 2011;10(5):475-85.
Edhemovic I, Brecelj E, Gasljevic G, Marolt Music M, Gorjup V, Mali B, et al. Intraoperative electrochemotherapy of colorectal liver metastases. J Surg Oncol. 2014;110(3):320-7.
de Almagro MC, Vucic D. Necroptosis: Pathway diversity and characteristics. Semin Cell Dev Biol. 2015;39:56-62.
Imamura H, Sakamoto S, Yoshida T, Matsui Y, Penuela S, Laird DW, et al. Single-cell dynamics of pannexin-1-facilitated programmed ATP loss during apoptosis. Elife. 2020;9.
Zhou J, Wang H, Wang W, Ma Z, Chi Z, Liu S. A Cationic Amphiphilic AIE Polymer for Mitochondrial Targeting and Imaging. Pharmaceutics. 2022;15(1).
Esser AT, Smith KC, Gowrishankar TR, Vasilkoski Z, Weaver JC. Mechanisms for the intracellular manipulation of organelles by conventional electroporation. Biophys J. 2010;98(11):2506-14.
Zhou C, Yan Z, Liu K. Response characteristics and optimization of electroporation: simulation based on finite element method. Electromagn Biol Med. 2021;40(3):321-37.
Bonferoni MC, Rassu G, Gavini E, Sorrenti M, Catenacci L, Torre ML, et al. Electrochemotherapy of Deep-Seated Tumors: State of Art and Perspectives as Possible "EPR Effect Enhancer" to Improve Cancer Nanomedicine Efficacy. Cancers (Basel). 2021;13(17).
Calvet, C. Y.; Famin, D.; Andre, F. M.; Mir, L. M. Electrochemotherapy with bleomycin induces hallmarks of immunogenic cell death in murine colon cancer cells. Oncoimmunology 2014, 3, e28131.
Reviewer comment 4: While this study claims the benefits that come along with ECT presenting it as an option for overcoming limitations such as heat-sink effect noted in RFA and MWA, the paper needs to acknowledge that other electroporation based therapies are also able to overcome these limitations. Please include how irreversible electroporation (IRE) is able to also overcome these effects, and discuss how this technique compares to ECT in the context of treating hepatic lesions. (Please see this review for a list of relevant literature https://pubmed.ncbi.nlm.nih.gov/28445252/)
Reply: According to the comment of the reviewer, we have included a novel paragraph in the Discussion section of our revised manuscript, which reads as follows:
‘Several ablative methods have been developed for the treatment of hepatic tumors, including irreversible EP, MWA, and RFA. In irreversible EP, membrane integrity is irreversible deranged through short pulses of high frequency direct current, inducing cell death (Nault et al., 2018). In contrast to rEP, irreversible EP leads to an abnormal transmembrane electrical potential across the cellular membrane causing permanent loss of cellular homeostasis (Lyu et al., 2017). Clinical studies in the treatment of primary or metastatic hepatic tumors reported unfavourable results following irreversible EP. Specifically, local recurrence free survival rate at one year was estimated at 59.5-70%, while increased incidence of recurrence was reported in tumors with diameter larger than 4 cm (Cannon et al., 2013; Sutter et al., 2017).‘
(See page 12, line 369-377)
References:
Nault JC, Sutter O, Nahon P, Ganne-Carrié N, Séror O. Percutaneous treatment of hepatocellular carcinoma: State of the art and innovations. J Hepatol. 2018;68(4):783-97.
Lyu T, Wang X, Su Z, Shangguan J, Sun C, Figini M, et al. Irreversible electroporation in primary and metastatic hepatic malignancies: A review. Medicine (Baltimore). 2017;96(17):e6386.
Cannon R, Ellis S, Hayes D, Narayanan G, Martin RC, 2nd. Safety and early efficacy of irreversible electroporation for hepatic tumors in proximity to vital structures. J Surg Oncol. 2013;107(5):544-9.
Sutter O, Calvo J, N'Kontchou G, Nault JC, Ourabia R, Nahon P, et al. Safety and Efficacy of Irreversible Electroporation for the Treatment of Hepatocellular Carcinoma Not Amenable to Thermal Ablation Techniques: A Retrospective Single-Center Case Series. Radiology. 2017;284(3):877-86.
Minor Revisions
Introduction
Reviewer comment 5: Introduction would benefit from an explanation of the field of electroporation-based modalities (i.e. deeper dive into reversible electroporation, followed by introduction of ECT)
Reply: According to the comment of the reviewer, we have included two novel paragraphs in the revised form of the manuscript, which reads as follows:
‘To overcome those limitations, new nonthermal ablation technologies have been developed based on the principle of electroporation (EP) of the cellular membrane through application of an external electric field. As EP is defined the phenomenon that occurs when cells are exposed to a high external electric field. Through EP, a transmembrane voltage is induced on the cellular membrane that exceeds a threshold value, causing formation of hydrophilic pores and increased cellular permeability. Electric pulses can cause reversible (rEP) or irreversible EP. In reversible EP, pore formation and membrane destabilization are transient, and cells regain homeostasis (Geboers et al., 2020). On the other hand, in the irreversible EP, magnitude and duration of applied electrical pulses overwhelm the adaptive capacity of cell membrane, causing irre-versible injury of cell membrane and cell death (Geboers et al., 2020; Savic et al., 2016).
Electrochemotherapy (ECT) represents an ablative procedure that combines the administration of chemotherapeutic agents with well-dosed electric pulses for cell membrane rEP. In ECT, the enhanced cellular permeability facilitates the transportation of otherwise poorly penetrating chemotherapeutic agents into tumor cells, increasing their cytotoxicity and inducing cell death (Probst et al., 2018; Edhemovic et al., 2014).’
(See page 2, lines 57-71)
References:
Geboers B, Scheffer HJ, Graybill PM, Ruarus AH, Nieuwenhuizen S, Puijk RS, et al. High-Voltage Electrical Pulses in Oncology: Irreversible Electroporation, Electrochemotherapy, Gene Electrotransfer, Electrofusion, and Electroimmunotherapy. Radiology. 2020;295(2):254-72.
Savic LJ, Chapiro J, Hamm B, Gebauer B, Collettini F. Irreversible Electroporation in Interventional Oncology: Where We Stand and Where We Go. Rofo. 2016;188(8):735-45.
Probst U, Fuhrmann I, Beyer L, Wiggermann P. Electrochemotherapy as a New Modality in Interventional Oncology: A Review. Technol Cancer Res Treat. 2018;17:1533033818785329.
Edhemovic I, Brecelj E, Gasljevic G, Marolt Music M, Gorjup V, Mali B, et al. Intraoperative electrochemotherapy of colorectal liver metastases. J Surg Oncol. 2014;110(3):320-7.
Reviewer comment 6: Line 54: It is mentioned that MWA and RFA are limited by tumor treatment size, alluding that ECT may be able to overcome this limitation. However, the literature suggests that 3cm is the maximum tumor diameter at which ECT treatment is efficacious. Please either include in next paragraph a reference that suggests that ECT can commonly treat larger tumor sizes, or remove this sentence as it suggests a false sense that ECT can overcome this limitation.
Reply: We agree with the reviewer that ECT has not been associated with beneficial outcomes in tumors larger than 3cm compared to RFA or MWA. Our statement in the Introduction section was slightly ambiguous. To make this clearer, we have rephrased our statement in the revised version of our manuscript, which now reads as follows:
‘Specifically, they are not indicated for tumors adjacent to major hepatic vessels or bile ducts due to the heat-sink effect, and tumors in the vicinity of other organs due to the risk of thermal injuries [1, 2, 4, 6-8].’
(See page 2, line 53-56)
Reviewer comment 7: It is necessary to mention the fact that reversible electroporation allows the pores formed from the applied field to reseal after a certain amount of time, alluding to its benefit of targeted drug delivery while the pores are ‘open’, then they seal with the drug inside.
Reply: This information is provided in the revised form of the manuscript in the Introduction section (see reply to comment 5).
Materials and Methods
Reviewer comment 8: Line 101: Please clarify if day 8 is 8 days after tumor inoculation. Was there a specific reason for waiting this length of time (wanted the tumor to grow to a desired size?)
Reply: The eighth day is 8 days after tumor implantation. The information is provided in the revised version of our manuscript (See page 3, line 103-104).
According to studies conducted with the same colon carcinoma cell line, 8-10 days after tumor implantation, a tumor with a diameter of 5-10 mm is developed in the liver lobe without extrahepatic metastases (Ziemann et al., 2019; Kauffels et al., 2019; Sperling et al., 2015; Sperling et al., 2013). Furthermore, within 8 days, tumor growth does not affect the animals systematically. Finally, in our study, the mean diameter of 5 mm was ideal for the electroporation of the tumor tissue with two parallel needle electrodes in a fixed geometry with interelectrode distance of 8 mm.
References:
Kauffels A, Kitzmuller M, Gruber A, Nowack H, Bohnenberger H, Spitzner M, et al. Hepatic arterial infusion of irinotecan and EmboCept® S results in high tumor concentration of SN-38 in a rat model of colorectal liver metastases. Clin Exp Metastasis. 2019;36(1):57-66.
Ziemann C, Roller J, Malter MM, Keller K, Kollmar O, Glanemann M, et al. Intra-arterial EmboCept S(R) and DC Bead(R) effectively inhibit tumor growth of colorectal rat liver metastases. BMC Cancer. 2019;19(1):938.
Sperling J, Ziemann C, Gittler A, Benz-Weisser A, Menger MD, Kollmar O. Tumour growth of colorectal rat liver metastases is inhibited by hepatic arterial infusion of the mTOR-inhibitor temsirolimus after portal branch ligation. Clin Exp Metastasis. 2015;32(4):313-21.
Sperling J, Brandhorst D, Schafer T, Ziemann C, Benz-Weisser A, Scheuer C, et al. Liver-directed chemotherapy of cetuximab and bevacizumab in combination with oxaliplatin is more effective to inhibit tumor growth of CC531 colorectal rat liver metastases than systemic chemotherapy. Clin Exp Metastasis. 2013;30(4):447-55.
Reviewer comment 9: The second paragraph of this section would flow better if it was placed after the Tumor cell implantation section and/or combined into section 2.3
Reply: In section 2.2, we describe the parts and the protocol of our study. For that reason, we included also all procedures after tumor cell implantation. After this section, we describe in sections 2.3, 2.4, 2.5, and 2.6 separately every conducted procedure, including tumor cell implantation, ECT, ultrasound, and blood cell analysis.
To make the experimental protocol clearer, we have rephrased the section 2.2, which now reads as follows:
‘The rats were randomized into four groups (n = 8 per group; 4 males and 4 females). All animals underwent laparotomy with tumor cell implantation in the left liver lobe (day 0). On day 8, relaparotomy was performed and all animals underwent ultrasound and photoacoustic imaging. Based on the treatment, the animals were divided in four groups. The first group underwent ECT with intravenous administration of bleomycin (BLM), the second group received only systemic chemotherapy with BLM, the third group underwent rEP with intravenous injection of an equivalent amount of 0.9% saline solution (B. Braun, Melsungen AG, Germany) and the fourth group underwent laparotomy and surgical exposure of the liver without treatment (Sham). BLM was used in the experiments, as it has been proven that among other tested chemotherapeutic agents BLM has the highest potentiation of cytotoxicity by rEP (up to 1,000 times) (References in manuscript).
Five days following the treatment (day 13), the animals underwent relaparotomy for the final ultrasound and photoacoustic imaging and for the collection of venous blood samples. Thereafter, the animals were sacrificed by an intravenous overdose of sodium pentobarbital and samples of the left liver lobe, including tumor and normal hepatic tissue, were asserved for histological and immunohistochemical analyses. The body weight of the animals was measured on days 0, 8 and 13 to assess eventual weight loss due to the treatment.’
(See page 3, lines 103-104)
Reviewer comment 12: Please specify where the CC531 cells obtained from?
Reply: The tumor cells were obtained from the ‘Cell Lines Service’. This information is provided in the revised version of the manuscript.
(See page 3, line 120)
Reviewer comment 11: Please indicate what is the motivation behind the selected pulse parameters?
Reply: In Sennex® Tumor-System, we do not have the option to change the pulse parameters. This ECT device was produced according to the European standardized protocol for the treatment of skin cancer (European Standard Operating Procedures of the Electrochemotherapy, ESOPE) (Gehl et al., 2018; Marty et al., 2006). ESOPE is a European multicenter non-randomized study that was conducted from the leading European cancer centers on ECT. Parameters associated with treatment effectiveness, such as number of treated nodules, size of tumor, number and type of electrodes, geometry of electrodes in tumor area, pulse parameters, and doses of chemotherapeutic agents have been defined according to the recommendations of the ESOPE.
Consequently, we did not have the option to change the pulse parameters in the Sennex® Tumor-System. Furthermore, the effect of different pulse parameters on the efficacy of the ECT treatment was out of the scope of this study.
According to the comment of the reviewer, we rephrased the second paragraph of the section 2.4 to indicate the rationale behind the utilization of the selected electrical parameters. The correction reads as follows:
‘According to the European guidelines for the utilization of ECT, the Sennex® Tumor-System provides a voltage between the pair of electrodes at 1.000 V, corresponding to an amplitude of 125 V/mm and frequency of 1 Hz (Gehl et al., 2018; Marty et al., 2006). ‘
(See page 3 and 4, lines 147-149)
References:
Gehl J, Sersa G, Matthiessen LW, Muir T, Soden D, Occhini A, et al. Updated standard operating procedures for electrochemotherapy of cutaneous tumours and skin metastases. Acta Oncol. 2018;57(7):874-82.
Marty M, Sersa G, Garbay JR, Gehl J, Collins CG, Snoj M, et al. Electrochemotherapy – An easy, highly effective and safe treatment of cutaneous and subcutaneous metastases: Results of ESOPE (European Standard Operating Procedures of Electrochemotherapy) study. European Journal of Cancer Supplements. 2006;4(11):3-13.
Reviewer comment 12: Can you further explain how the pulse generator provided feedback on treatment progress/feedback? How is it that the generator is able to determine an incomplete
therapy?
Reply: We contacted the BIONMED® Technologies, which produces the ECT Sennex® Tumor-System. The feedback of the device is based on the galvanic element, which is utilized as an indicator of the tendency to sufficient electroporation. However, the device cannot define if the electroporation of the tumor and the electric field coverage were sufficient.
For that reason and to avoid a misinterpretation of our statement, we removed this sentence in the section 2.4 of Materials and Methods.
Reviewer comment 13: What was the electrode exposure?
Reply: The electrode tip exposure is 5mm.
Reviewer comment 14: How can you obtain a 100 V/mm voltage to distance ratio if you are applying 1000 V across an 8 mm spacing. This would be 125 V/mm. Please correct.
Reply: The reviewer is correct that the amplitude is estimated at 1250 V/cm. We contacted the company BIONMED®Technologies GmbH, which produces the ECT device (Sennex® Tumor-System). They confirmed that the voltage between the pair of electrodes was set to 1.000 V, corresponding to an amplitude of 1250 V/cm (125 V/mm) and frequency of 1 Hz. The amplitude has been corrected in the revised form of the manuscript.
(See page 3 and 4, line 147-149)
Reviewer comment 15: How was this voltage amplitude determined? It is in line with the same amplitude used for Irreversible electroporation (1000 + V/cm).
Reply: As reported in the reply of the comment 11, the amplitude was determined from the Sennex® Tumor-System and there is not an option for the user to change that. The pulse parameters were developed according to the European standardized protocol for the treatment of skin cancer (European Standard Operating Procedures of the Electrochemotherapy, ESOPE) (Gehl et al., 2018; Marty et al., 2006) (see reply to comment 11). The ESOPE protocol with eight square wave pulses of 100 μs uses charging voltages of 100–1,000 V to trigger reversible electroporation in the 0.6–1.5 kV/cm electric field range.
In irreversible EP, more pulses (at least 80-100 pulses) and a higher amplitude (up to 3,000 V) are required (Geboers et al., 2020). The most clinical trials adopt a voltage amplitude of 1,500–1,800 V/cm and current of 20-50 A as the treatment parameters. The pulse duration for clinical trials is 70-100 ms, most commonly 90 or 100 ms. Usually, the efficacy is achieved after a treatment of 90 pulses (Zhang et al., 2021).
References:
Geboers B, Scheffer HJ, Graybill PM, Ruarus AH, Nieuwenhuizen S, Puijk RS, et al. High-Voltage Electrical Pulses in Oncology: Irreversible Electroporation, Electrochemotherapy, Gene Electrotransfer, Electrofusion, and Electroimmunotherapy. Radiology. 2020;295(2):254-72.
Zhang N, Li Z, Han X, Zhu Z, Li Z, Zhao Y, et al. Irreversible Electroporation: An Emerging Immunomodulatory Therapy on Solid Tumors. Front Immunol. 2021;12:811726.
Reviewer comment 16: The last sentence in this section is a bit awkward. It should be rephrased to increase fluidity of reading.
Reply: According to the comment of the reviewer, the last sentence in section 2.5 of Materials and Methods was rephrased and reads as follows:
‘All ultrasound images were examined independently by three researchers with experience in liver ultrasound imaging. The final evaluation for each animal was given after agreement of the three researchers.’
(See page 4, lines 185-187)
Reviewer comment 17: Please indicate in your methodology what the blood samples will be used for and what key information will be extracted from this.
Reply: According to the comment of the reviewer, we rephrased the section 2.6 of Materials and Methods, which reads as follows:
‘The blood cell analysis was conducted to evaluate the inflammatory reaction and the blood loss following the treatment.’
(See page 5, lines 190-191)
Results
Reviewer comment 18: Please further specify how vascularization was studied? What specific cells or morphologies were examined?
Reply: Tumor vascularization was examined with the platelet endothelial cell adhesion molecule-1 (PECAM-1; CD31). PECAM-1 is expressed exclusively on endothelial cells and can function as an indicator for vascularization.
Further information on cells and morphologies, which were examined, is given in the revised form of the manuscript. The section 3.6 of the Results reads as follows:
‘Tumor vascularization was examined with the identification of CD31-positive blood vessels. CD31 is expressed exclusively on the endothelial cells of blood vessels and can utilized as an indicator of tissue vascularization.’
(See page 9, lines 324-326)
Reviewer comment 19: Why would there be an increase in inflammatory response in the sham group?
Reply: The inflammatory response in all groups was clarified in the reply of the comment 3a. In the immunohistological analysis, we analysed the MPO-positive neutrophilic granulocytes. The main function of granulocytes is the phagocytosis of the malignant cells. Since the Sham group presented minimal necrosis of the tumor tissue, it was normal that the infiltration of granulocytes will be increased in this group. Although the rEP and BLM groups had the same levels of tumor necrosis with the Sham group, they presented lower levels of inflammatory cells. However, there was also a trend of increased infiltration of inflammatory cells in the rEP and BLM groups.
Discussion
Reviewer comment 20: Line 370: what is ‘by ca. 400%’?
Reply: We thank the reviewer for this information. The reviewer is correct that the effectiveness is not increased by 400% in the ECT group. The necrosis was estimated at 20.0 ± 3.1% in the BLM group, whereas in the ECT was estimated at 87.2 ± 3.1%. That means an increase in the effectiveness of BLM by ca. 335% in the ECT group.
The correct information is provided in the revised version of the manuscript.
(See page 13, line 402)
Reviewer comment 21: Lines 384-385: the ‘preclinical studies’ referenced here should be cited.
Reply: The correction is provided in the revised form of the manuscript.
Reviewer comment 22: Lines 393-394: the ‘in vitro studies’ referenced here should be cited.
Reply: The correction is provided in the revised form of the manuscript.
Reviewer comment 23: Lines 423-429: If electric fields induce vascular lock, how are you seeing increased microvascularization and increased hemoglobin and oxygenation in rEP? Is it then the fields that are inducing the lock, or the presence of the drug?
Reply: The vascular lock is characterized by vascular contraction. EP increases the antitumoral efficacy of the treatment through vascular contraction and hypoperfusion, inducing drug entrapment. Application of electric field is associated with electrical-induced reflexive constriction of arterioles and interstitial edema of endothelial structures. Consequently, decreased blood flow in the treated areal increases the retaining time of the chemotherapeutic agent and prevents bleeding during intervention (Probst et al., 2018). For that reason, the results of vascular lock have been demonstrated only in the ECT group, where EP was combined with BLM. In the rEP group, the vascular contraction and tumor hypoperfusion following the apply of the electrical field are transient without affecting the hemoglobin concentration and oxygenation of the tumor tissue.
Figures
Reviewer comment 24: In most figures, the letter assignments are difficult to see. In Figure 2, suggestion to move ‘I’ and ‘J’ to be outside of the plot to the left of the figures. In Figures 3-7, either make the letters A-D larger, brighter in a different color, or similarly move them to the left of the figure. Do the same for the ‘E’ plots (move the letter to the side).
Reviewer comment 25: Bar plots in figure 2 are a bit blurry. Please upload a higher resolution image.
Reply to comment 24 and 25: According to the comment of the reviewer, we corrected the figures in the revised form of the manuscript.

Reviewer 3 Report
Electrochemotherapy is a widely used method for the treatment of superficial (skin) tumors, but with limited application for internal lesions. The presented work is interesting and methodologically well organized. I have some remarks and questions for the authors:
1. What is the shape of the applied pulses? Is the duration of the applied pulses too long?
2. Is the device battery powered or connected to the mains during pulse application?
3. Is the application of electrical pulses with an electrode with a fixed interelectrode distance of 8 mm appropriate for tumor sizes of 5 mm in diameter? Healthy surrounding tissue was also electroporated. There is a difference between tumor tissue and healthy liver tissue that can be the cause of necrosis. On the other hand you have apoptosis in the Sham group but not in the ECT group. Please provide more explanation for these results.
4. Line 138 mentions that there is feedback on the pulse feed. What parameters do you follow as feedback?
Author Response
Review of the manuscript ID ‘cancers-2184116’ by Spiliotis et al.
Reply to the comments of reviewer 3
Reviewer comment 1: What is the shape of the applied pulses? Is the duration of the applied pulses too long?
Reply: Square shaped electric pulses with short duration of 100 μs were applied in the tumor tissue.
Reviewer comment 2: Is the device battery powered or connected to the mains during pulse application?
Reply: The Sennex® Tumor-System is not battery powered. The device is connected to the mains during the pulse application.
Reviewer comment 3: Is the application of electrical pulses with an electrode with a fixed interelectrode distance of 8 mm appropriate for tumor sizes of 5 mm in diameter? Healthy surrounding tissue was also electroporated. There is a difference between tumor tissue and healthy liver tissue that can be the cause of necrosis. On the other hand you have apoptosis in the Sham group but not in the ECT group. Please provide more explanation for these results.
Reply: In ECT, the electrodes are inserted around the tumor within the healthy tissue (Geboers et al., 2020; Probst et al., 2018). In this manner, an electric field that is distributed in the whole tumor tissue can be developed. Depending on the tumor size, more than one EPs can be performed in order to achieve adequate tumor coverage with the electric field. In our study, the small size of the tumors allowed us to use the electrodes with interelectrode distance of 8mm to achieve sufficient EP of the malignant cells. However, analyses of the produced electric field with those pulse parameters and electrodes geometry have not been conducted, since that was not within the scope of our study.
As you can see in the following figure (unpublished data from the BIONMED® Technologies GmbH), the electric field is sufficient to achieve coverage of the tumor tissue. In the figure is demonstrated the electric field with 4 electrodes. Because of the small tumor size, we utilized only two electrodes.
Normal hepatocytes adjacent to electrodes were also electroporated. However, necrosis was observed at the tumor tissue that had been exposed to the electrical pulses, whereas normal hepatic tissue between electrodes and tumor was not affected. An ablated area was detected at the electrodes’ insertion site, which was oval with a diameter of 3-4 mm. In these foci, the hepatic tissue was necrotic, including perforated blood vessels or dilated capillaries with RBC extravasates, and was surrounded by inflammatory cells.
The fact that the electroporated normal hepatic tissue remains unaffected is attributed to the mechanism of action of rEP in combination with BLM. BLM predominantly affects actively dividing cancer cells at the stage of mitosis (cell cycle G2) and to a lesser extent the normal non-dividing cells in the surrounding normal hepatic tissue (Miklavčič et al., 2012). Therefore, cytotoxic effects are observed in the tumor tissue, where the electric field is applied, without affecting the normal hepatic parenchyma. The negligible damage of normal hepatic tissue after ECT had been confirmed in preclinical studies. Specifically, necrosis was observed at the treated tumor tissue, whereas minimal histological changes with well-defined coagulation necrosis were detected around the inserted electrodes at the normal hepatic tissue (Zmuc et al., 2019; Jaroszeski et al., 2001). The necrosis around the electrodes is attributed to thermal injury or irreversible EP of the hepatocytes and is limited only to cells adjacent to electrodes.
Regarding the increased number of apoptotic cells in the Sham group, we have provided a detailed clarification in the comment 3a of the reviewer 2. Please see reply of the comment 3a.
This information is provided in the revised version of the manuscript.
(See page 14, lines 438-452)
References:
Miklavčič D, Serša G, Brecelj E, Gehl J, Soden D, Bianchi G, et al. Electrochemotherapy: technological advancements for efficient electroporation-based treatment of internal tumors. Med Biol Eng Comput. 2012;50(12):1213-25.
Zmuc J, Gasljevic G, Sersa G, Edhemovic I, Boc N, Seliskar A, et al. Large Liver Blood Vessels and Bile Ducts Are Not Damaged by Electrochemotherapy with Bleomycin in Pigs. Sci Rep. 2019;9(1):3649.
Jaroszeski MJ, Coppola D, Nesmith G, Pottinger C, Hyacinthe M, Benson K, et al. Effects of electrochemotherapy with bleomycin on normal liver tissue in a rat model. Eur J Cancer. 2001;37(3):414-21.
Reviewer comment 4: Line 138 mentions that there is feedback on the pulse feed. What parameters do you follow as feedback?
Reply: We contacted the BIONMED® Technologies, which produces the ECT Sennex® Tumor-System. The feedback of the device is based on the galvanic element, which is utilized as an indicator of the tendency to sufficient electroporation. However, the device cannot define if the electroporation of the tumor and the electric field coverage were sufficient.
For that reason and to avoid a misinterpretation of our statement, we removed this sentence in the section 2.4 of Materials and Methods.

Reviewer 4 Report
I suggest the Authors to introduce information regarding the distribution of the electric field in the tumoral tissue. As they use 2 needles only the electric field may not cover the whole tumour.
Author Response
Review of the manuscript ID ‘cancers-2184116’ by Spiliotis et al.
Reply to the comments of reviewer 4
Reviewer comment: I suggest the Authors to introduce information regarding the distribution of the electric field in the tumoral tissue. As they use 2 needles only the electric field may not cover the whole tumour.
Reply: In our study, the small size of the tumors allowed us to use the electrodes with interelectrode distance of 8mm to achieve sufficient EP of the malignant cells. However, analyses of the produced electric field with those pulse parameters and electrodes geometry have not been conducted, since that was not within the scope of our study. Consequently, we cannot provide data of electric field distribution on the tumor tissue. Furthermore, since the produced electric field depends on pulse parameters, electrodes geometry and tissue characteristics, we cannot report in the Discussion section data of other studies, which have used other electric parameters and other tissues.
As you can see in the following figure (unpublished data from the BIONMED® Technologies GmbH), the electric field is sufficient to achieve coverage of the tumor tissue. In the figure is demonstrated the electric field with 4 electrodes. Because of the small tumor size, we utilized only two electrodes.

Round 2
Reviewer 3 Report
I accept the chages.